# Structure of the G$_q$-coupled adhesion receptor ADGRL4

Qingchao Chen[1], Anastasiia Gusach ●[1], Aurora Diamante[2], Jayesh C. Patel[2], Patricia C. Edwards[1], Christopher G. Tate ●[1] & David M. Favara ●[1,3,4] ✉

Adhesion G protein-coupled receptors (aGPCRs) are a 32-member family of Class B GPCRs that have diverse cellular roles including mechanosensation, cell-fate determination, neurodevelopment, immune function and tumour biology. ADGRL4 is upregulated in the tumour microenvironment and is implicated in tumour pathogenesis across a broad range of malignancies. Inhibiting ADGRL4 is a potential therapeutic treatment for currently intractable cancers such as glioblastoma. Previous work suggested that ADGRL4 does not signal through G protein coupled pathways. However, using a sensitive bioluminescent assay, we demonstrate here that ADGRL4 couples weakly to the heterotrimeric G protein G$_q$, whilst there is no robust coupling to other G proteins (G$_s$, G$_{12}$, G$_o$) or β-arrestin 1 or 2. We determine the cryo-EM structure of ADGRL4 coupled to a heterotrimeric G$_q$ complex to a resolution of 3.1 Å. The overall fold of ADGRL4 is similar to that of other aGPCRs, but the coupling to G$_q$ is distinct with fewer interactions between the receptor and G protein. The structure is consistent with ADGRL4 being activated by its tethered agonist and represents an important step towards the development of potential inhibitors for the treatment of multiple tumour types.

G protein-coupled receptors (GPCRs) form the largest receptor family in the human proteome, orchestrating a multitude of physiological processes[1]. Among them are adhesion GPCRs (aGPCRs), a 32-member GPCR family that are characterised by their large extracellular domains, which facilitate cell-cell and cell-matrix interactions, and play a diverse range of biological roles in cell-fate determination, mechanosensation, immune function, neurodevelopment, metabolism, and tumour biology[2]. Most aGPCRs contain a GPCR autoproteolysis-inducing (GAIN) domain immediately before the first transmembrane helix (Fig. 1a), which possesses a hidden tethered agonist also known as the stachel[3–6]. During biosynthesis in the endoplasmic reticulum the GAIN domain undergoes autoproteolysis[7], cleaving the receptor into an N-terminal fragment (NTF; most of the N-terminal domain) and a C-terminal fragment (CTF; all of the transmembrane regions), which remain non-covalently associated as the receptor is trafficked to the cell membrane[8]. Upon ligand binding or

mechanical disruption of the NTF-CTF interaction, the NTF is released to expose the tethered agonist which subsequently activates the receptor by binding to its orthosteric binding pocket (Fig. 1a). This activation model is supported by recently determined cryo-EM structures of active-state aGPCRs (in order of structure determination: ADGRG3, ADGRG1, ADGRL3, ADGRD1, ADGRG5, ADGRG2, ADGRG4, ADGRF1, ADGRE5)[9–19].

ADGRL4 (ELTD1) is a highly conserved aGPCR[20] implicated in the pathogenesis of a broad range of malignancies[21], including (but not limited to) glioblastoma[22–25], gastric cancer[26,27], colorectal cancer[28–30], breast cancer[31], renal cancer[28,32], ovarian cancer[28], hepatocellular carcinoma[33,34], cholangiocarcinoma[35], retinoblastoma[36] and neuroblastoma[37]. In general, higher ADGRL4 expression within the tumour microenvironment across the majority of these tumours correlates with more aggressive disease in vitro and in vivo[23,24,26,27,29,30,32,34,36,37]. Mechanistically, ADGRL4 drives angiogenesis[28,38], proliferation and

[1]MRC Laboratory of Molecular Biology, Francis Crick Avenue, Cambridge, UK. [2]Nxera Pharma UK Limited, Steinmetz Building, Granta Park, Great Abington, Cambridge, UK. [3]Department of Oncology, University of Cambridge, Cambridge, UK. [4]Department of Oncology, Addenbrookes' Hospital, Cambridge University Hospitals NHS Foundation Trust, Cambridge, UK. ✉e-mail: dfavara@mrc-lmb.cam.ac.uk

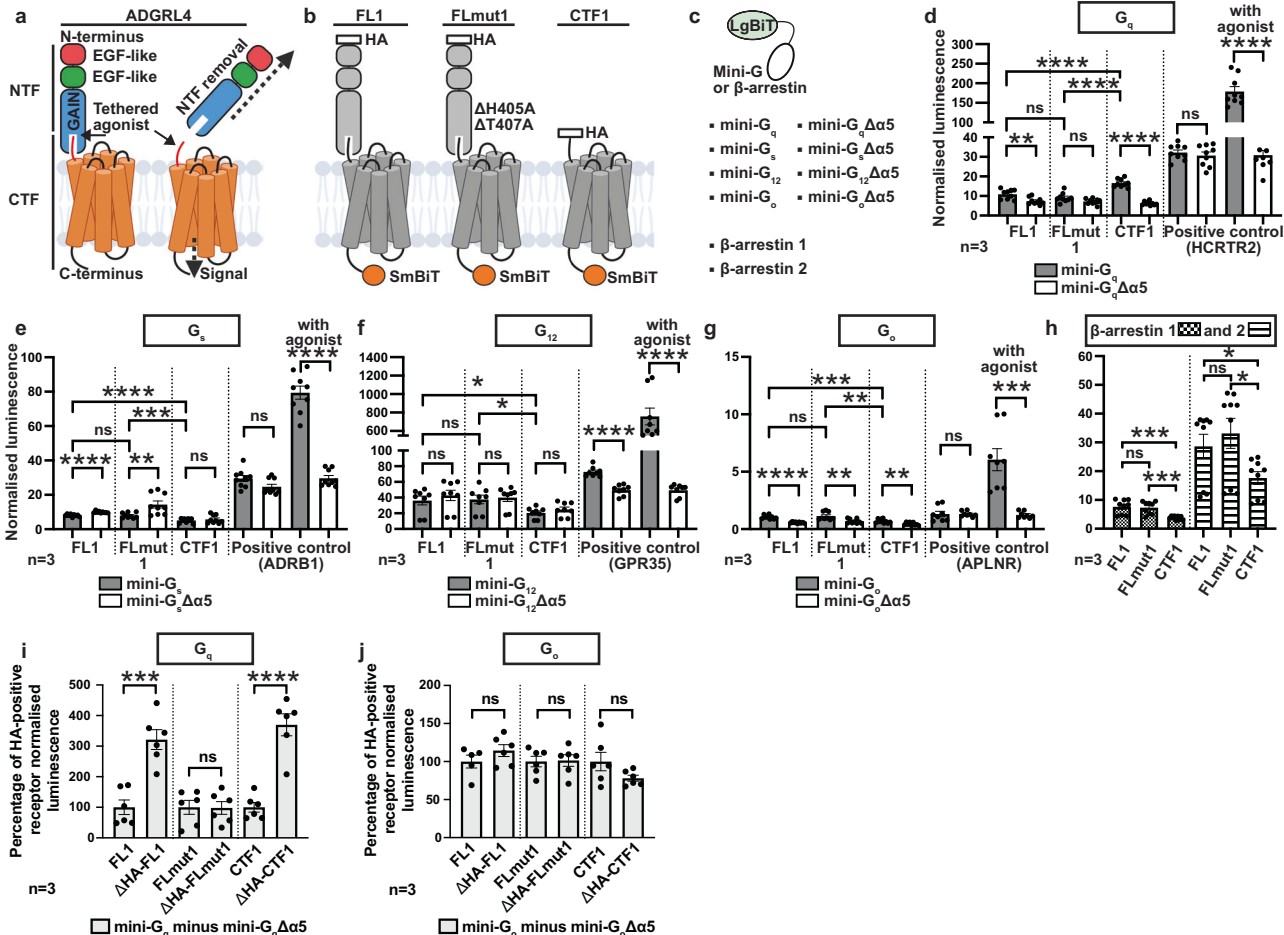

**Fig. 1 | ADGRL4 couples weakly to mini-$G_q$. a** Cartoon representation of ADGRL4 highlighting its extracellular domains and the activation mechanism. Created in BioRender. Favara, D. (2026) https:// BioRender.com/82k5188. **b** Cartoon representation of ADGRL4 coupling constructs. **c** Cartoon representation of N-terminal LgBiT tethered mini-G protein or β-arrestin constructs. **d** Interactions between constructs of ADGRL4-SmBiT and LgBiT-mini-$G_q$ (grey bars) were determined by luminescence and compared to cells transfected with ADGRL4-SmBiT constructs and LgBiT-mini-$G_q\Delta\alpha5$ controls (white bars). Luminescence results were normalised by the expression levels of SmBiT- and LgBiT-tagged constructs (Supplementary Fig. 1). Data represent three biological replicates, with each assay containing 3 technical replicates; all comparisons were tested using an unpaired two-tailed Student's t-test; error bars represent the SEM (****, $p < 0.0001$; ***, $p < 0.001$; **, $p < 0.01$; *, $p < 0.05$; ns, non-significant). **e–g** Interactions between constructs of ADGRL4-SmBiT and a variety of LgBiT-mini-G proteins (grey bars) or LgBiT-mini-G$\Delta\alpha5$ control proteins (white bars): **e** mini-$G_s$; **f** mini-$G_{12}$; **g** mini-$G_o$. Data represents three biological replicates with each assay containing three technical replicates; all comparisons were tested using an unpaired two-tailed Student's t-test; error bars represent the SEM. **h** Interactions between constructs of ADGRL4-SmBiT and either LgBiT-βarr1 (chequered bars) or LgBiT-βarr2 (lined bars). Data represents three biological replicates with each assay containing three technical replicates; all comparisons were tested using an unpaired two-tailed Student's t-test; error bars represent the SEM (***, $p < 0.001$; *, $p < 0.05$; ns, non-significant).

**i**, Assessment of the effect of N-terminal HA tags on $G_q$ coupling. Interactions between aGPCR constructs (-SmBiT tagged) and LgBiT-mini-$G_q$ or LgBiT-mini-$G_q\Delta\alpha5$ constructs were determined by luminescence. Luminescence results were normalised by the expression levels of SmBiT- and LgBiT-tagged constructs. LgBiT-mini-$G_q\Delta\alpha5$ normalised luminescence samples were subtracted from LgBiT-mini-$G_q$ samples, with results presented as a percentage of HA-positive constructs. ΔHA versions represent receptors without N-terminal HA tags. Data represent three biological replicates, with each assay containing two technical replicates; all comparisons were tested using an unpaired two-tailed Student's t-test; error bars represent the SEM (****, $p < 0.0001$; ***, $p < 0.001$; **, $p < 0.01$; *, $p < 0.05$; ns, non-significant). **j** Assessment of the effect of N-terminal HA tags on $G_o$ coupling. Interactions between aGPCR constructs (-SmBiT tagged) and LgBiT-mini-$G_o$ or LgBiT-mini-$G_o\Delta\alpha5$ constructs were determined by luminescence. Luminescence results were normalised by the expression levels of SmBiT- and LgBiT-tagged constructs. LgBiT-mini-$G_o\Delta\alpha5$ normalised luminescence samples were subtracted from LgBiT-mini-$G_o$ samples, with results presented as a percentage of HA-positive constructs. ΔHA versions represent receptors without N-terminal HA tags. Data represent three biological replicates, with each assay containing two technical replicates; all comparisons were tested using an unpaired two-tailed Student's t-test; error bars represent the SEM (****, $p < 0.0001$; ***, $p < 0.001$; **, $p < 0.01$; *, $p < 0.05$; ns, non-significant). All data and significance values for graphs **d–j** are shown in Supplementary Table 1.

migration[23,26,29,37], epithelial-mesenchymal transition (EMT)[26,39,40] and regulates endothelial metabolism[41]. Several in vivo systems indicate ADGRL4 is a promising therapeutic target[28,42–45], although no synthetic and no commercial therapeutic agents targeting ADGRL4 are yet available or under clinical trial. Much remains unknown about this receptor, including its activation, signalling mechanism(s) and protein structure. Past comprehensive signalling screens have concluded that ADGRL4 does not couple to G proteins[38,46], whilst studies in cancer cell lines have shown it to activate downstream components of the JAK/STAT3[23] and

MAPK/ERK[26,37] pathways; both pathways can be triggered via multiple mechanisms, including G protein coupling. Resolving ADGRL4's activation mechanism and mapping its precise mode of signalling has broad implications for understanding its role in tumour biology and for designing targeted antagonists.

Here, we present evidence for the direct coupling of a G protein to ADGRL4 and describe the high-resolution cryo-EM structure of its active-state conformation, offering mechanistic insights into how an aGPCR implicated in multiple cancers is activated and how it could be

pharmacologically targeted. Establishing this receptor's direct G protein coupling not only clarifies a longstanding debate in the aGPCR field but also offers an alternative sensitive approach to investigating other aGPCRs previously reported as not coupling to G proteins.

## Results

### ADGRL4 couples to $G_q$

Previous studies of GPCR coupling have concluded that ADGRL4 does not couple to G proteins[38,46]. We hypothesised that this conclusion could have resulted from insufficient assay sensitivity rather than an intrinsic inability to couple. To address this, we employed the sensitive SmBiT/LgBiT NanoLuc complementation reporter system[47] to evaluate G protein coupling. In brief, this system comprises two complementary fragments of NanoLuc luciferase (a 1.3-kDa SmBiT fragment and a larger 18-kDa LgBiT fragment) tethered to the proteins under investigation: upon protein-protein interaction, the fragments reassemble to form an active luciferase which, in the presence of substrate, provides a quantitative readout of the interaction dynamics.

Three G protein coupling ADGRL4 constructs were designed (Fig. 1b). The ADGRL4 construct 'Full-Length 1' (FL1) was the human wild-type full-length protein (amino acid residues 2-690). 'Mutated Full-Length 1' (FLmut1) was the full-length construct containing the alanine substitutions H405A and T407A at the GPCR proteolysis site (GPS), which prevented autocatalytic cleavage and hence NTF and CTF dissociation. Finally, the 'C-terminal Fragment 1' (CTF1) was a truncated construct encoding ADGRL4 from the GPS cleavage site to its C-terminal end (amino acid residues 407-690). All of the constructs contained an N-terminal HA tag and a C-terminal SmBiT complementation fragment tethered via a 12-amino acid GS linker. The FLmut1 construct was hypothesised to be predominantly inactive (due to blocked NTF/CTF dissociation) whilst the CTF1 construct was hypothesised to be constitutively active through interaction of the tethered agonist with the orthosteric binding pocket.

The G protein-LgBiT constructs[48] comprised mini-G proteins[49,50] or β-arrestin variants tethered to N-terminal LgBiT via a 15-amino acid GS linker (Fig. 1c). Four different G protein-LgBiT constructs were tested (LgBiT-mini-$G_q$, LgBiT-mini-$G_s$, LgBiT-mini-$G_{12}$, LgBiT-mini-$G_o$) and two β-arrestin constructs were also tested (LgBiT-βarr1 and LgBiT-βarr2). Mini-$G\alpha 5$ versions of each mini-G variant were used as negative controls for mini-G complementation experiments. These versions lacked part of the α5 helix critical for GPCR coupling[51,52], thus preventing any interaction.

HEK293T cells were transiently transfected with receptor-SmBiT and LgBiT-mini-G/mini-$G\alpha 5$ or LgBiT-βarr constructs and assayed 48 hours after transfection. Cell surface expression of HA-tagged SmBiT-tethered constructs was evaluated, as was whole-cell expression of both SmBiT- and LgBiT-tethered constructs (Supplementary Fig. 1a-c). Representative mean cell surface expression of CTF1 (5%) was significantly lower than for FL1 and FLmut1 (88%) (Supplementary Fig. 1a, and Source Data file) suggesting either rapid cell-surface internalisation of activated receptor, expression and activity at subcellular locations, or protein misfolding. This difference was not seen for whole-cell expression where representative results showed marginally higher expression for CTF1 than FL1 and FLmut1.

Results show that ADGRL4-CTF1, the constitutively active mutant, coupled weakly with mini-$G_q$ when compared to a positive control, the orexin receptor 2 (HCRTR2), a known $G_q$-coupled receptor[53] (Fig. 1d). Importantly, the coupling of ADGRL4-CTF1 to mini-$G_q$ was significantly greater than binding to the negative control, mini-$G_q\Delta\alpha 5$ ($p < 0.0001$) (Fig. 1d; Supplementary Fig. 2a; Supplementary Table 1). As expected, the $G_q$-coupling activity of ADGRL4-CTF1 was significantly greater than the coupling observed to either the wild type receptor, ADGRL4-FL1, or to the inactive mutant receptor ADGRL4-FLmut1 ($p < 0.0001$) (Fig. 1d; Supplementary Fig. 2a; Supplementary Table 1). No robust coupling was detected to mini-$G_s$, mini-$G_{12}$, mini-$G_o$, βarr1 and βarr2 (Fig. 1e-h;

Supplementary Fig. 2a-b; Supplementary Table 1) corroborating previous findings[38,46]. We noted that the data for mini-$G_o$ was potentially consistent with coupling, but as it was so weak and the inactive mutant FLmut1 appeared to couple with $G_o$ more than the constitutively active mutant CTF1, we did not pursue this further (Fig. 1g; Supplementary Fig. 2a; Supplementary Table 1).

As N-terminal tags may affect GPCR coupling and signalling, we investigated whether N-terminal HA tags negatively affected our G protein coupling results. Constructs without an N-terminal HA tag were therefore developed (ΔHA-FL1; ΔHA-FLmut1, ΔHA-CTF1) and were assayed with the same NanoLuc assay. Results for mini-$G_q$ confirm our finding that ADGRL4 couples to $G_q$, albeit with lower efficiency when an N-terminal HA-tag is present on the tethered agonist (Fig. 1i, Supplementary Table 1). N-terminal HA tagging apparently had no effect on coupling with mini-$G_o$ (Fig. 1j, Supplementary Table 1). As we required the N-terminal HA tag for our structure pipeline, namely for detection during cell line development and during purification, we continued to make use of HA-tagged constructs.

### Cryo-EM structure determination of the ADGRL4·mini-$G_q$·βγ complex

Four ADGRL4 constructs were designed for cryo-EM studies and differed from the previous constructs used in the G protein coupling assays only by their purification tags, fluorescent reporter fusions and mini-$G_q$ fusions (Supplementary Fig. 3a). All purification constructs contained the same N-terminal tags (HA, twin Strep, His10 and eGFP tags), separated by protease cleavage sites (HRV 3 C, TEV). The naming nomenclature is the same except that '1' is replaced by '2'. Thus ADGRL4-FLmut2 is the uncleavable full-length receptor (analogous to FLmut1) and ADGRL4-CTF2 is the constitutively active truncated receptor, analogous to ADGRL4-CTF1. The construct ADGRL4-CTF2A is analogous to ADGRL4-CTF2, but has C-terminally tethered mini-$G_q$ (version A), corresponded to the mini-$G_q$ used for G protein coupling experiments that does not bind βγ. ADGRL4-CTF2B is identical to ADGRL4-CTF2 except that the C-terminally tethered mini-$G_q$ is 'version B' that forms a complex with the βγ-subunits (see Methods).

All four constructs for cryo-EM study were stably expressed in inducible HEK293 GnTi⁻ TetR cells (Supplementary Fig. 3b-e). ADGRL4-FLmut2 had the highest level of whole-cell and cell-surface expression, whilst ADGRL4-CTF2 had the lowest with close to no surface expression suggesting either instability, misfolding and/or rapid turnover of the potentially constitutively active receptor (Supplementary Fig. 3d,e). Interestingly, the tethering of C-terminal mini-$G_q$ (version A or B) to ADGRL4-CTF2 (generating CTF2A and CTF2B respectively) substantially improved whole-cell and cell-surface expression, suggesting increased receptor stability once tethered with mini-$G_q$, presumably due to constitutive coupling of the mini-$G_q$ to ADGRL4 (Supplementary Fig. 3d,e). In contrast to the other constructs, overexpressing ADGRL4-CTF2B for 72 hours led to significant cell death (Supplementary Fig. 3d), presumably resulting from its constitutive activity and ability of its tethered mini-$G_q$ (version B) to bind βγ, resulting in increased activity via βγ. Forty-eight hours was optimal for maximal expression of most of the constructs (Supplementary Fig. 3e) and fluorescence-detection size-exclusion chromatography (FSEC) showed that both DDM-CHS and LMNG-CHS were both suitable for purification and structure determination (Supplementary Fig. 3f,g). Although ADGRL4-CTF2A had better surface expression, larger FSEC peaks, and induced less cell death than ADGRL4-CTF2B (Supplementary Fig. 3d-f), we ultimately selected ADGRL4-CTF2B as the most suitable candidate for cryo-EM studies. This decision was based on the ability of its tethered mini-$G_q$ (version B) to bind βγ, thus increasing the molecular weight of the ADGRL4-$G_q$ complex and serving as a good fiducial marker to facilitate particle alignment during cryo-EM image processing. ADGRL4-CTF2B was therefore purified in LMNG-CHS (see Methods), purified βγ added and the complex vitrified on cryo-EM

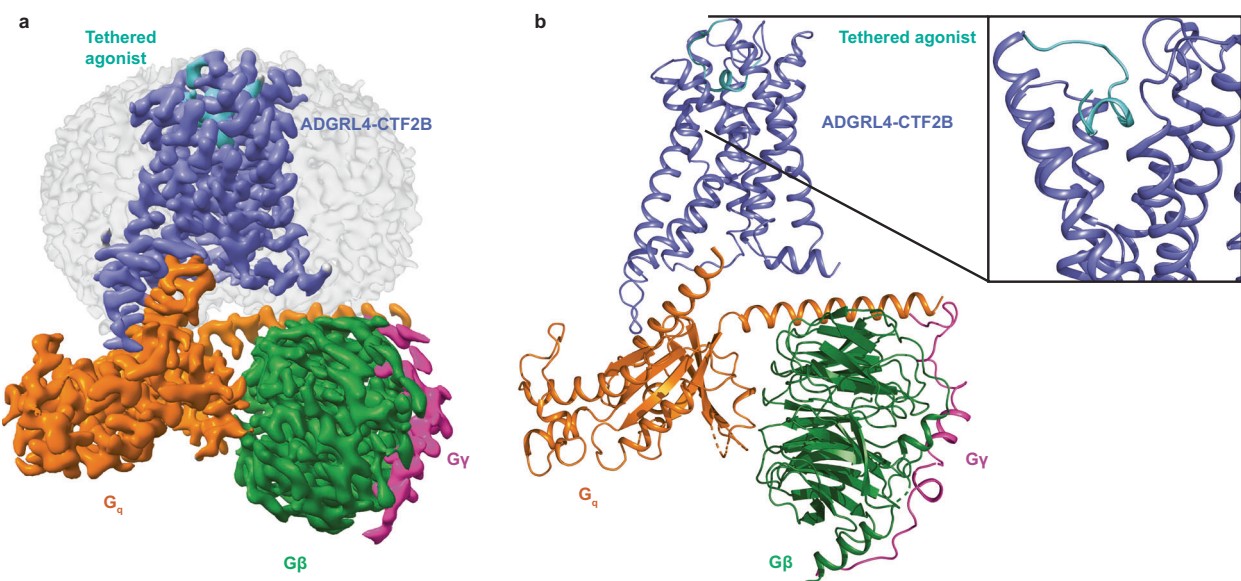

**Fig. 2 | 3.14 Å Cryo-EM structure of active-state ADGRL4-CTF2B. a, b** Cryo-EM density map (**a**) and structure model (**b**) for active-state ADGRL4-CTF2B in complex with mini-$G_q$ and βγ. The tethered agonist is highlighted in turquoise with the insert showing an alternative magnified view.

grids (Supplementary Fig. 3b,h-j). The structure was subsequently determined of the ADGRL4-CTF2B-βγ complex to an overall nominal resolution of 3.1 Å (Fig. 2a,b, Supplementary Fig. 4a-f and Supplementary Table 2). For convenience, from here onwards the tethered mini-$G_q$ (version B) bound to the βγ-subunits will be referred to as $G_q$, as done previously for the majority of other aGPCR-G protein structures[9,11,12,14–16,18,19].

The overall structure of ADGRL4-CTF2B is consistent with the canonical fold of GPCRs and the structures of previously determined aGPCRs[9–19,54]. The tethered agonist region is folded into a short α-helix that binds within the orthosteric binding pocket and makes key interactions with transmembrane helices H1, H2, H3, H5, H6 and H7 (Fig. 3a-c). The tethered $G_q$ couples to ADGRL4-CTF2B in a manner consistent with a Class B2 GPCR, forming key contacts between the α5 helix of $G_q$ and ADGRL4's transmembrane helices (H2, H3, H5, H6) and intracellular loop (ICL) 3 (Fig. 3d).

### Structural comparison of ADGRL4-$G_q$ to other $G_q$-coupled aGPCRs

The structure of the ADGRL4-$G_q$ complex was compared to previously published structures of aGPCRs coupled to $G_q$, namely ADGRL3[16], ADGRE5[18] and ADGRF1[14] (Fig. 4a-c). The overall transmembrane region structures are conserved with ADGRL4 being most similar to ADGRL3 and ADGRE5 (in both cases RMSDs are 1.0 Å) and less similar to ADGRF1 (RMSD 1.9 Å). Regions of the receptors that show the greatest similarity are the transmembrane domains and the position of the short helix within the tethered agonist (Fig. 4a-c). This suggests that the overall activation pathway for the receptors is highly conserved, although there will undoubtedly be subtleties in the kinetics of activation that cannot be deduced solely from structures. The greatest differences between the structures are observed in the extracellular loops (ECL1,2 and 3) and the intracellular loops (ICL1,2 and 3) along with the position of the cytoplasmic ends of H5 and H6.

A comparison of amino acid contacts (within ≤3.9 Å) formed between the tethered agonist and receptor in the orthosteric binding pocket shows that only one contact point (Thr[1.43]) is fully conserved across all four $G_q$-coupled active-state aGPCRs (Fig. 4c). Two additional contact points (Phe[2.64] and Phe[7.42]) are conserved among three of the receptors (ADGRL4, ADGRL3, ADGRE5); superscripts refer to the universal numbering scheme from[55]. In the ECL2 region, although not identical across all four structures, two to three hydrophobic residues

(either Val, Leu, Tyr, Phe or Trp) make contacts to the tethered agonist, with only a Trp residue being conserved in three out of the four receptors and making contact to the tethered agonist helical region.

To identify the side-chains critical for ADGRL4 activation, alanine scanning mutagenesis was performed on eight tethered agonist residues (T407[B.GPS.+1]-S415[B.S14.53]; GAIN generic residue number scheme[56] in superscript) forming contacts (≤3.9 Å) with residues within the orthosteric binding pocket, as determined by our cryo-EM structure. Mutations were introduced into the constitutively active ADGRL4-CTF1 G protein coupling construct (C-terminally SmBiT-tagged) and assessed for $G_q$ coupling using the NanoLuc complementation assay. Seven of the eight substitutions significantly reduced $G_q$ coupling, with six (H408A[B.S14.46], F409A[B.S14.47], I411A[B.S14.49], L412A[B.S14.50], M413A[B.S14.51], S414A[B.S14.52]) reducing $G_q$ activity by >50% of wild-type CTF1. M413A[B.S14.51] abolished activity entirely ($p < 0.0001$) and F409A[B.S14.47] produced the second largest decrease to 12% of wild-type CTF1 ($p < 0.0001$) (Fig. 3e, Supplementary Fig. 5a, Supplementary Table 3). These findings are consistent with the cryo-EM structure and identify that the most important amino acids in the tethered agonist region for ADGRL4 activation are M413[B.S14.51] followed by F409[B.S14.47] within the hydrophobic block F409-x-I411-L412-M413 (where x is residue A410 which does not form any contacts in our cryo-EM model).

Published aGPCR structures have proposed three putative core activation motifs within the seven transmembrane (7TM) core: 1) the upper quaternary core motif (UQC; F[3.44]; M[3.47]; F[5.43]/F[5.47]; W[6.53])(Wootten residue numbering[57]) as reported for ADGRG2 and ADGRG3[11–13,58] (W[6.53] has a hydrogen bond with Q[7.49]); 2) the hydrophobic P/F/W/LφφG motif (where φ represents a hydrophobic residue) as reported for ADGRF1 and ADGRD1[11,13,14]; and 3) the H(N)L(M)Y motif as reported in ADGRG3 and ADGRD1[11,13,58]. These are summarised in the review by Seufert et al[59]. We assessed whether ADGRL4 contained any of these putative 7TM activation motifs by performing sequence alignment against all human protein-coding aGPCRs and by comparing the relative residue positions in the abovementioned aGPCRs with associated canonical 7TM activation motifs, with the structure of ADGRL4. Results (Fig. 5a, Supplementary Fig. 6) show that all three putative 7TM core activation motifs are present in ADGRL4, albeit with some differences. The UQC motif in ADGRL4 comprises F505[3.44], M508[3.47] and W631[6.53], with a hydrogen bond between W631[6.53] and Q656[7.49]. The F[5.43]/F[5.47] component of the canonical UQC motif is absent in ADGRL4. The P/F/W/LφφG motif in ADGRL4 comprises F625[6.47], L626[6.48], L627[6.49], and

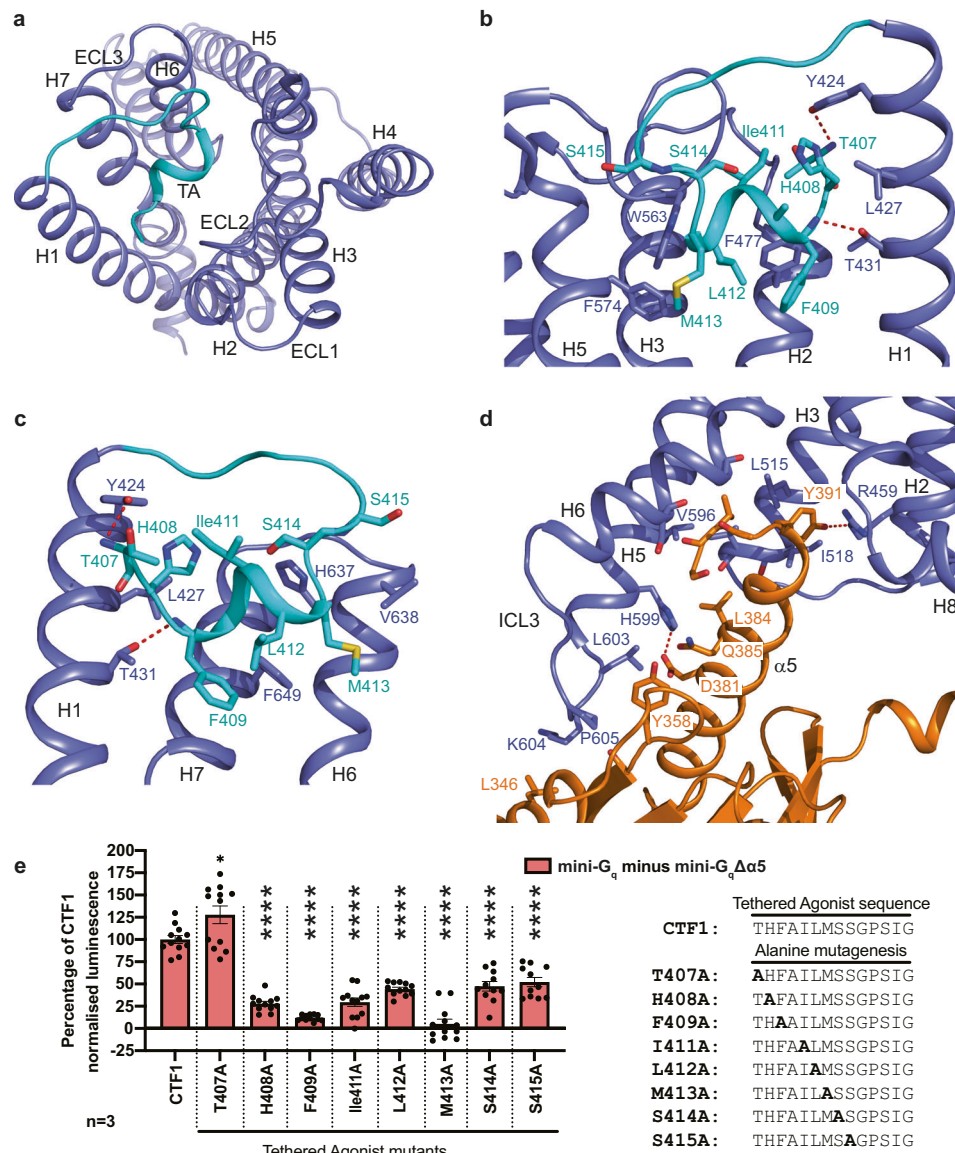

**Fig. 3 | Active-state ADGRL4's tethered agonist peptide interactions and interactions between ADGRL4 and G$_q$. a** Global ribbon view of ADGRL4's tethered agonist in relation to all transmembrane helices. **b, c** Magnified view of the interactions between the tethered agonist and residues in H1, H2, and H3 (**b**); and H1, H6 and H7 (**c**). The bound tethered agonist is coloured in turquoise; dotted red lines represent potential hydrogen bonds. **d** Detailed view of the interactions between the α5 helix of G$_q$ with H2, H3, H5, H6 and ICL3 of ADGRL4. G$_q$ is depicted in orange, ADGRL4 in purple (TA: tethered agonist). **e** Alanine mutagenesis performed on amino acid residues within the tethered agonist region which formed contacts with the receptor's orthosteric binding pocket within ≤3.9 Å. Interactions between

mutagenesis constructs (-SmBiT tagged) and LgBiT-mini-G$_q$ or LgBiT-mini-G$_q$Δα5 constructs were determined by luminescence. Luminescence results were normalised by the expression levels of SmBiT- and LgBiT-tagged constructs. LgBiT-mini-G$_q$Δα5 normalised luminescence samples were subtracted from LgBiT-mini-G$_q$ samples, with results presented as a percentage of CTF1 G$_q$ activity. Data represent three biological replicates, with each assay containing four technical replicates; comparisons were tested using an unpaired two-tailed Student's t-test; error bars represent the SEM (****, $p < 0.0001$; *, $p < 0.05$). All the data values and significance values for graph **e** are shown in Supplementary Table 3.

G628[6.50] (FφφG); and its H(N)L(M)Y motif comprises H514[3.53], L515[3.54] and Y516[3.55]. When comparing putative activation motifs across all aGPCRs, the UQC motif appears to have the highest level of sequence conservation suggesting functional relevance (Fig. 5a).

Alanine scanning mutagenesis was performed on all three putative 7TM activation motifs to interrogate functional effect (a total of 11 mutants were generated). Mutations were introduced into the constitutively active ADGRL4-CTF1 construct with G$_q$ coupling assessed using the NanoLuc complementation assay. All mutants were expressed similarly to wild type ADGRL4-CTF1 as determined by cell surface anti-HA flow cytometry (Supplementary Fig. 5b). All the mutants exhibited a significant decrease in G$_q$ coupling when compared to wild

type L4-CTF (Fig. 5b, Supplementary Fig. 5c, Supplementary Table 4). Mutation of any of the amino acid residues in the UGC abolished G$_q$ recruitment. Mutations of two amino acid positions in each of the FφφG and H(N)L(M)Y motifs (L627A[6.49] and G628A[6.50], and L515A[3.54] and Y516A[3.55], respectively) also abolished G$_q$ recruitment, although slightly lesser effects where observed at residues F625A[6.47] and L626A[6.48] in the FφφG motif and for H514A[3.53] in the H(N)L(M)Y motif. (Fig. 5b, Supplementary Fig. 5c, Supplementary Table 4). In summary, these results provide the evidence that ADGRL4 contains all three 7TM activation motifs (Fig. 5c) and that these all play an important role in ADGRL4 activation and function, with loss of the UQC having the greatest negative effect.

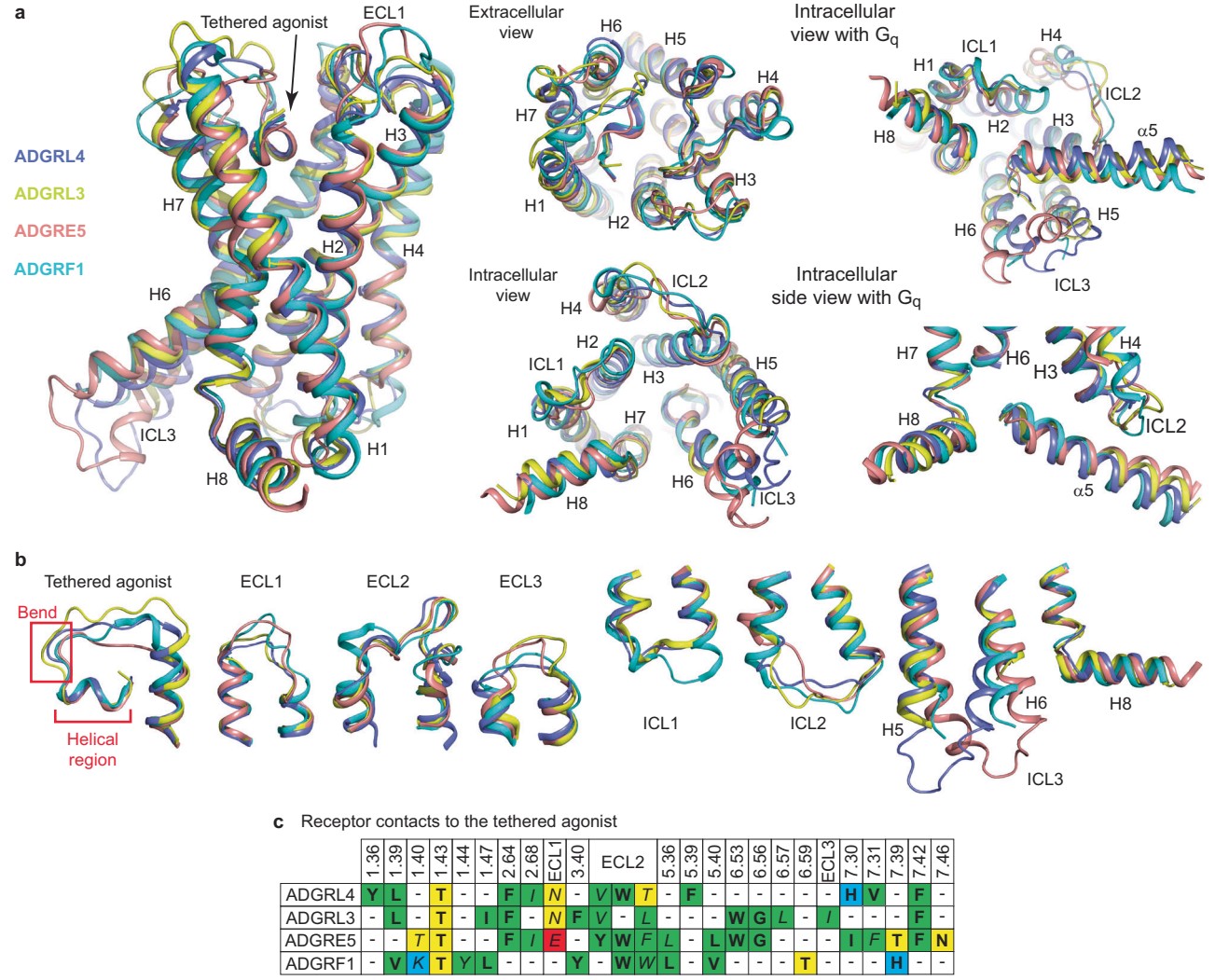

**Fig. 4 | Comparison of the active-state structures of $G_q$-coupled ADGRL4, ADGRL3, ADGRE5 and ADGRF1. a** Structural superposition of ADGRL4 (purple), ADGRL3 (yellow), ADGRE5 (orange) and ADGRF1 (green). **b** Diversity of loop structures between receptors; the coloration is the same as in (**a**). **c** Comparison of the amino acid contacts (distance cut-off ≤3.9 Å) made by the tethered agonist and receptor within the orthosteric binding pocket for ADGRL4, ADGRL3, ADGRE5 and ADGRF1 coupled to $G_q$: bold, those residues making contacts to the helical region; italics, those residues making contacts to the bend; green, hydrophobic residues; blue, positively charged residues; red, negatively charged residues; yellow, polar uncharged residues. PDB IDs: ADGRL3, 7wy5; ADGRE5, 7ydm; and ADGRF1, 7wxu.

Contacts between the four aGPCRs and $G_q$ are highly variable (Fig. 6a-c) and there are only two amino acid residues in all four receptors that always make a contact (positions 3.57 and 5.61; Ile518 and Val596 in ICL3 of ADGRL4) (Fig. 6b). H3 and H5 are the only two transmembrane α-helices that have contacts to $G_q$ in all four receptors. What is unique about the pattern of contacts in ADGRL4 is that no contacts are made to $G_q$ by residues in ICL2 and that there are contacts made by ICL3 not observed in any of the other $G_q$-coupled aGPCR structures (Fig. 6b). In two receptors (ADGRL3 and ADGRF1), ICL3 is disordered, suggesting that if there are contacts to the G protein, they are transient and weak. ICL3 in ADGRE5 is ordered but does not contact $G_q$. The cytoplasmic end of transmembrane H6 in ADGRL4 is tilted by 30° compared to that in ADGRE5, resulting in a 7 Å shift to the end of the α-helix and hence a different position of ICL3 (Fig. 4b, Fig. 6b).

There are marked differences in the regions of $G_q$ that make contact to the receptors and also in the overall number of contacts made, which will impact the strength of interaction between the aGPCR and $G_q$ (Fig. 6c). For example, in ADGRL4 there are only a few weak interactions with the N-terminal region of the α5 helix (H5.08-H5.22; G protein nomenclature from[60]) whereas seven residues in this region of $G_q$ make contacts to ADGRL3. In contrast, there are strong interactions made between ICL3 of ADGRL4 and an α−loop−β region of $G_q$ (H4.12, h4s6.20, S6.01, S6.03) that are absent in the ADGRL3-$G_q$ structure. The β-subunit of $G_q$ makes strong polar interactions with ADGRL3 that are absent in the ADGRL4-$G_q$ complex. The number of atoms in $G_q$ that are within 3.9 Å of the receptor is also considerably fewer in the ADGRL4-$G_q$ complex compared to other aGPCR-$G_q$ complexes, ranging between 38%-54% of the number observed in the $G_q$ complexes formed by ADGRL3, ADGRE5 and ADGRF1. In addition, there are three favourable charge-charge interactions between ADGRL3 and $G_q$ involving Lys/Arg residues and Glu residues, and a further five potential hydrogen bonds at the interface (Fig. 6a). In contrast, there is only one potential hydrogen bond at the interface between ADGRL4 and $G_q$, and no favourable charge-charge interactions (Fig. 6a). These observations are all consistent with our initial G protein coupling data, which indicated that ADGRL4 binds $G_q$ relatively weakly.

## Discussion

In this work we demonstrate that ADGRL4, an adhesion GPCR implicated in the pathogenesis of multiple malignancies, couples weakly to $G_q$, and by employing a tethered mini-$G_q$ strategy to stabilise ADGRL4

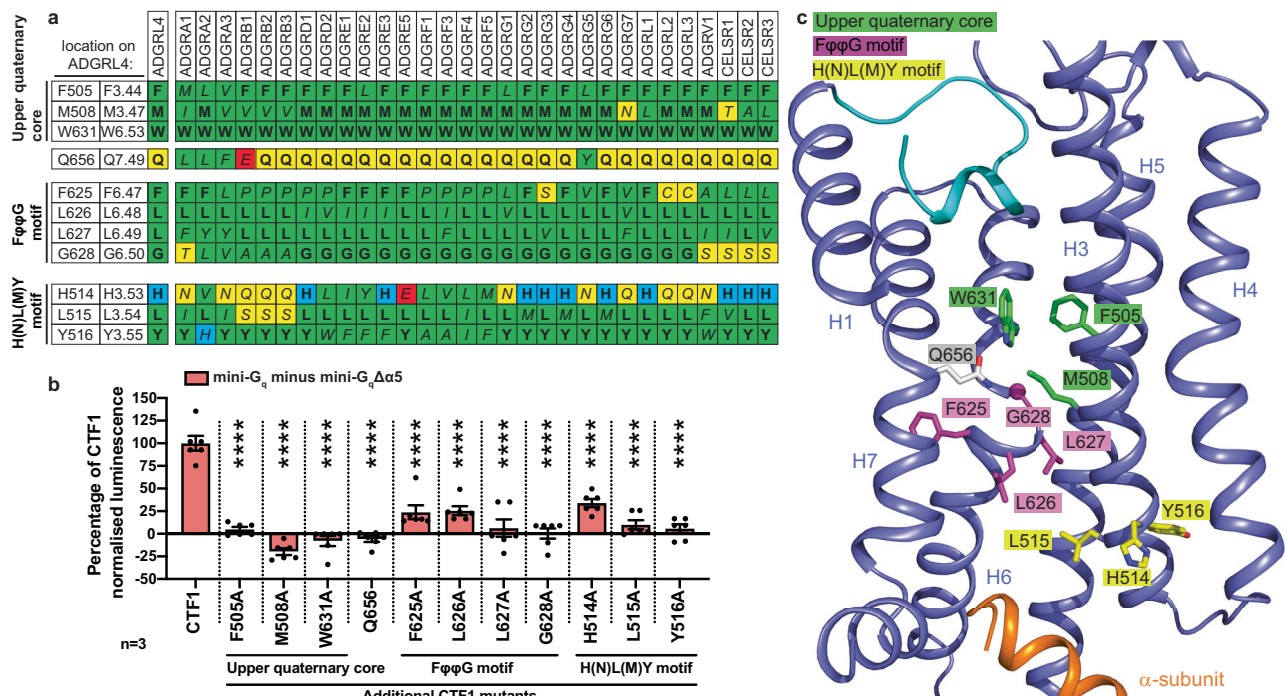

**Fig. 5 | Activation motifs in ADGRL4. a** Amino acid residue conservation in aGPCRs of the upper quaternary core (UQC), FφφG motif and H(N)L(M)Y motif regions (φ refers a hydrophobic residue in this context): bold, conserved residues; italics, non-conserved residues; green, hydrophobic residues; blue, positively charged residues; red, negatively charged residues; yellow, polar uncharged residues. **b** Alanine mutagenesis performed on amino acid residues within three putative 7TM activation motifs (UQC, FφφG motif and H(N)L(M)Y motif regions) with interactions between mutagenesis constructs (-SmBiT tagged) and LgBiT-mini-$G_q$ or LgBiT-mini-$G_q\Delta\alpha5$ constructs determined by luminescence. Luminescence results were normalised by the expression levels of SmBiT- and LgBiT-tagged constructs. LgBiT-mini-$G_q\Delta\alpha5$ normalised luminescence samples were subtracted from LgBiT-mini-$G_q$ samples, with results presented as a percentage of CTF1 $G_q$ activity. Data represent three biological replicates, with each assay containing two technical replicates; comparisons were tested using an unpaired two-tailed Student's t-test; error bars represent the SEM (****, $p < 0.0001$). All data and significance values for graph **b** are shown in Supplementary Table 4. **c** Structure of ADGRL4 (purple) with activation motif residues indicated: green, UQC motif; magenta, FφφG motif; yellow, H(N)L(M)Y motif. Q656[7.49] (forming a hydrogen bond with W631[6.53]) is listed in grey. Parts of H1 and H2 have been removed for clarity. The tethered agonist (cyan) and the C-terminus of $G_q$ (orange) are also shown.

in its active state, reveal a cryo-EM structure of active-state ADGRL4 at an overall resolution of 3.1 Å. Our G protein coupling results challenge previous screening studies (which also incorporated both FL and CTF ADGRL4 variants)[38,46] which reported that ADGRL4 does not engage heterotrimeric G proteins. ADGRL4's weak $G_q$ coupling was likely overlooked in earlier assays that lacked sufficient sensitivity to detect lower-affinity interactions between aGPCR and G protein. Our bioluminescence proximity-based tethered-mini-G approach to coupling thus offers an alternative sensitive approach to investigating the remaining eleven aGPCRs reported as not coupling to G proteins or activated by their predicted tethered agonist sequence[46]. If these remaining receptors are indeed shown to couple weakly to a G protein, employing a tethered approach such as ours could aid resolving their active-state structures. Whilst this paper was under review, another group has corroborated our $G_q$ findings by means of an NFAT-reporter assay (downstream of $G_q$) in U87 glioblastoma cells[61]. This suggests that cellular context plays an important role as previous ADGRL4 NFAT-reporter studies in HEK293 cells were negative[38,46]. Using a secreted HaloTag fusion of ADGRL4's extracellular domains as bait, the authors also affinity-purified proteins from U87-glioblastoma-cell-line-conditioned medium, and by mass spectrometry identified the first candidate ADGRL4 ligands: Ku80 (a nuclear DNA-repair protein) and erythrocyte β-spectrin (a cytoskeletal protein)[61].

The ADGRL4-$G_q$ structure highlights the key role played by the tethered agonist in ADGRL4's activation, mirroring observations noted in other active-state aGPCR structures[9–19,54]. This insight clarifies how ADGRL4 is activated: first, removal of the NTF exposes the tethered agonist; once the tethered agonist has been freed, it is able to bind the

orthosteric binding pocket, triggering the hallmark outward displacement of helix 6 (H6) on the receptor's cytoplasmic side which creates a cleft for $G_q$ engagement. Notably, ADGRL4 exhibited fewer contacts with mini-$G_q$ relative to other active-state $G_q$-coupled aGPCRs, consistent with the weak coupling observed in our G protein coupling assays. Our alanine mutagenesis results show that seven residues within ADGRL4's tethered agonist region contribute to activation (H408[B.S14.46], F409[B.S14.47], I411[B.S14.49], L412[B.S14.50], M413[B.S14.51], S414[B.S14.52], S415[B.S14.53]), the most critical being the hydrophobic M413[B.S14.51] and F409[B.S14.47] residues. These results largely conform to the canonical FXφφφXφ motif noted in other aGPCRs[11,12,62] (X=any residue; φ = a hydrophobic residue such as Ile, Leu and Met), except that ADGRL4 contains a polar serine (S415[B.S14.53]) at the final φ position.

Our investigation of putative 7TM activation motifs within ADGRL4 shows that all three previously reported hydrophobic activation motifs (UQC, FφφG and H(N)L(M)Y motifs) are present in ADGRL4 and are functionally important for ADGRL4 activation, with mutagenesis of the UQC motif residues having the largest negative effect on receptor activity. The UQC motif, canonically defined as F[3.44], M[3.47], F[5.43]/F[5.47], and W[6.53], is proposed to stabilise the transmembrane helix bundle in active conformations of ADGRG2 and ADGRG3[11–13,58]. In our ADGRL4 structure, three of the four canonical UQC residues were conserved, including W[6.53], which is stabilised by a hydrogen bond interaction with Q[7.49] as is observed in the canonical UQC motif. The F[5.43]/F[5.47] component of the UQC motif is absent in ADGRL4, as it is in other aGPCRs such as ADGRF1, ADGRL3, and ADGRG5, suggesting that this position may be dispensable or variable across aGPCR subfamilies. The conserved FφφG in ADGRL4 mediates the transmembrane helix 6

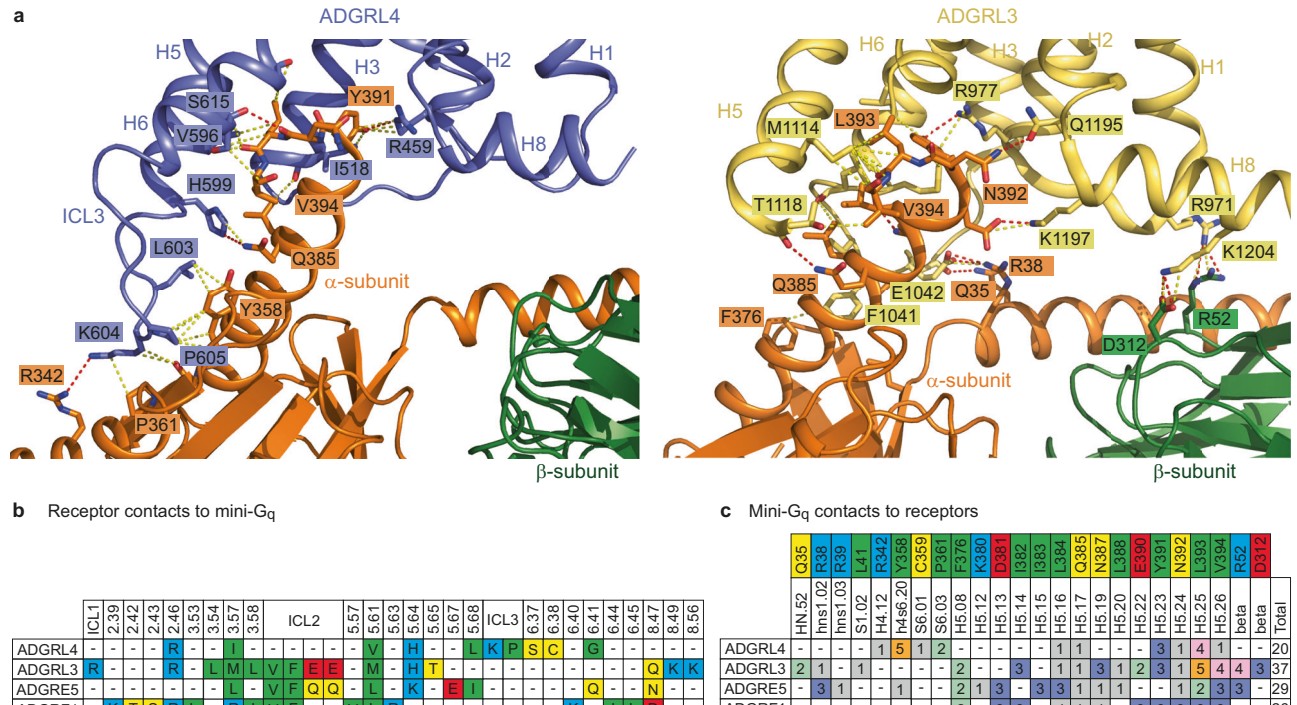

**Fig. 6 | Comparison of receptor-mini-G$_q$ interfaces. a** Interactions between ADGRL4 (turquoise) or ADGRL3 (yellow; PDB ID: 7wy5) with the α-subunit (orange) and the β-subunit (dark green) of the mini-G$_q$ heterotrimeric G protein are depicted. Structures are shown as cartoons with residues at the interfaces depicted as sticks (red, oxygen atoms; dark blue, nitrogen atoms). Dashes represent interatomic interactions: yellow, van der Waals interactions (distance ≤ 3.9 Å); red, potential polar contacts. **b** Comparison of the amino acid contacts (distance cut-off ≤3.9 Å) made by the receptor to mini-G$_q$ for active-state ADGRL4, ADGRL3, ADGRE5 and ADGRF1; green, hydrophobic residues; blue, positively charged residues; red, negatively charged residues; yellow, polar uncharged residues. PDB IDs: ADGRL3-mini-G$_q$, 7wy5; ADGRE5-mini-G$_q$, 7ydm; and ADGRF1-mini-G$_q$, 7wxu. **c** Comparison between the amino acid contacts made by mini-G$_q$ to receptor for active-state ADGRL4, ADGRL3, ADGRE5 and ADGRF1. Residue colouring scheme as before; contact frequency coloured separately.

kink. Importantly, Q$^{7.49}$, which interacts with W$^{6.53}$, is also closely packed against the FφφG motif, implying an analogous structural role in stabilizing TM6 conformation during activation. In the H(N)L(M)Y motif of ADGRL4 (H$^{3.53}$, L$^{3.54}$, Y$^{3.55}$), L$^{3.54}$ directly contacts Y391 of the α5 helix at the C-terminus of G$_q$, forming a hydrophobic interface with the G protein. In summary, our results show that ADGRL4 contains all three 7TM activation motifs with partially conserved spatial architecture, and with all three motifs being important for receptor activation.

From a functional perspective, our results reconcile the literature linking ADGRL4 to downstream JAK/STAT3[23] and MAPK/ERK[26,37] activation in various cancer cell lines. In these experiments, ADGRL4 overexpression in a glioblastoma cell line activated JAK/STAT3 signalling, whereas ADGRL4 knockdown attenuated it[23]. Similarly, ADGRL4 overexpression in a gastric cancer cell line induced MAPK/ERK, whilst knockdown suppressed activation[26]. Finally, silencing ADGRL4 in a neuroblastoma cell line inhibited ERK/STAT3 signalling[37]. Whilst both JAK/STAT3 and MAPK/ERK pathways can be driven by a range of upstream pathways (for example tyrosine kinases, integrins, G proteins or paracrine factors), our results present a plausible mechanism where upon NTF release and tethered agonist exposure, ADGRL4 couples to G$_q$ and activates a branch of the MAPK/ERK or JAK/STAT3 signalling pathways, dependent on the cellular context. This insight offers a mechanistic framework explaining how G$_q$-coupled ADGRL4 could drive functionally significant downstream pathways.

A central motivation for this work lies in the frequent observation of overexpressed ADGRL4 in high-grade unresectable tumours such as glioblastoma, which carry limited therapeutic options and dismal survival rates. There is increasing in vitro and in vivo evidence demonstrating that antagonising ADGRL4 expression or activity can prolong survival[23,42,43,45,63]. Despite these clear links, no clinically tested or FDA approved antagonists currently exist, presenting an opportunity for therapeutic development. The structure of ADGRL4 thus represents a significant step forward in the development of novel tool compounds that inhibit it. This is essential to provide a proof of concept for future therapeutic interventions, which might be either nanobodies[64], synthetic binders[65] or small molecules[64]. It is highly desirable that only ADGRL4 is targeted and clearly the differences in ECL1-3 structure between ADGRL4 and other aGPCRs are a prerequisite for the successful development of subtype specific nanobodies or synthetic binders. They also may provide regions that could offer differences in binding of small molecules that also encompass these regions[66]. Of course, the structure of ADGRL4 determined here is in the active state and ideally a structure in the inactive state would be used for future drug discovery efforts. Work towards this goal is in progress, but the current structure now facilitates building a model of the inactive state based on the template of ADGRL3 in an inactive state[67]. The utility of models for in silico drug discovery has been recently demonstrated for the serotonin receptor[68]. The ability of structures to provide templates for the computational design of synthetic binders has also recently been proven[65] and, combined with ultra-high throughput screening methodologies[69] provide new routes to rapid development of targeted binders. A recent study identified Tanshinone IIA as an agent that could reduce ADGRL4 protein levels and potentially interact with it[34] and could serve as a viable lead in structure-based efforts to therapeutically target ADGRL4.

In terms of limitations, we used engineered mini-G chimeras for measuring the recruitment of G protein to the receptor, and used a C-terminally tethered mini-G$_q$ for structure determination. Mini-G proteins lack the G$_α$ helical domain and have been shown to form high-affinity, stable nucleotide-insensitive receptor complexes for

structural biology applications[49,50]. Our mini-$G_q$ coupling sensor was a $G_s$/$G_q$ chimera[49] in which residues in the α5 helix were mutated to match $G_q$. This has the potential to subtly alter coupling in comparison to wild type G proteins.

Another potential limitation is that the NanoBiT G protein recruitment assay reports proximity between SmBiT-tagged receptor and LgBiT-tagged mini-G rather than assessing GDP-GTP exchange, GTP hydrolysis or downstream second-messenger production, and it was performed using overexpressed ADGRL4 in HEK293T cells. The dominant $G_q$ result derives from a constitutively active CTF with markedly lower surface expression than full-length constructs. We detected no robust coupling to other transducers in this background, although a very weak trend with mini-$G_o$ was observed. We acknowledge that determination of physiologically relevant G protein coupling must be demonstrated in native endothelial cells or tumours. The intention of this paper was to determine an active-state structure of ADGRL4, and we needed to have a stable receptor-G protein complex to facilitate this work.

Finally, our conclusions on the structure and G protein recruitment assays should be viewed in the context of the requirements for ADGRL4 expression, purification and structure determination. The structure represents an active-state C-terminal fragment (CTF2B) of ADGRL4 bound to a C-terminally tethered mini-$G_q$ chimera that recruits βγ, assembled and analysed in LMNG/CHS detergent micelles with N-terminal tags for purification and detection retained. These choices were necessary to stabilise the weakly coupled ADGRL4-$G_q$ complex in vitro and to facilitate purification and alignment during cryo-EM image processing, but they could conceivably bias receptor-G protein geometry and orientation, omit potential membrane-lipid allostery, and mask alternative or transient states. Single-particle cryo-EM emphasises the most populated conformation, so it remains possible that low-occupancy active-state conformations or dynamic states exist that were not recovered by our classification as our analysis relied on averaging the most frequent 3D-conformations. A potential criticism that can be made of the ADGRL4 structure is that the presence of the N-terminal tags has affected the structure in some way. We do not think this is the case given the close similarity in position of the tethered ligand in ADGRL4 with other aGPCRs and also the positions of conserved amino acid residue side chains in the conserved motifs of aGPCRs. Of course, structures present a static view of a protein in a thermodynamically favourable state in the conditions used for structure determination and will not yield information regarding, for example, the kinetics of activation. Thus the N-terminal tags have a demonstrable effect on the ability of ADGRL4 to recruit $G_q$ (Fig. 1i), yet the structure is of relevance in understanding how ADGRL4 functions.

In summary, by demonstrating $G_q$ coupling for ADGRL4 and determining its first active-state high resolution structure, we reconcile prior negative results and provide a structural foundation for future anticancer drug discovery studies. Future studies will focus on generating antagonists with the ultimate aim of improving survival outcomes in the malignancies in which ADGRL4 plays a role.

## Methods

### G protein coupling construct design
**SmBiT-complementation coupling constructs.** All three G protein coupling codon-optimised ADGRL4 constructs were designed and cloned into a pcDNA 3.1 expression vector for coupling studies using standard cloning techniques. All constructs included an N-terminal haemagglutinin (HA) epitope tag for detection and a C-terminal SmBiT complementation fragment tethered via a 12-amino acid GS linker. Using the Uniprot ADGRL4 reference sequence (Q9HBW9), untagged Full-Length ADGRL4 constructs comprised amino acids K2-R690, whilst untagged C-terminal Fragment constructs comprised amino acids T407-R690.

**LgBiT-complementation constructs.** LgBiT constructs comprising mini-G protein[48–50] or β-arrestin[48] variants tethered to LgBiT via a 15-amino acid GS linker were used: 1) $G_q$: LgBiT-mini-$G_q$ (using 'mini-$G_{s/q}$ version 70'[49], a mini-G chimera comprising a backbone of 'mini-$G_s$ version 393'[49] with all residues which contact the GPCR altered to be the same as $G_q$'[49]; 2) $G_s$: LgBiT-mini-$G_s$ (using 'mini-$G_s$ version 399'[49]); 3) $G_{12}$: LgBiT-mini-$G_{12}$ (using 'mini-$G_{12}$ version 8'[49]); 4) $G_o$: LgBiT-mini-$G_o$ (using 'mini-$G_o$ version 17'); 5) β-arrestin 1: LgBiT-βarr1; 6) β-arrestin 2: LgBiT-βarr2 (as previously described[48]). As negative controls, mini-$G\alpha5$ versions for each mini-G protein variant above were used: all had a partially deleted α5 helix that prevented GPCR coupling[51,52].

### G protein coupling assays
Low passage mycoplasma-free HEK293T cells (ATCC CRL-3216; authenticated by ATCC) were transiently transfected with versions of receptor-SmBiT and LgBiT-mini-G/mini-$G\alpha5$ or LgBiT-β-arrestin, using GeneJuice (Sigma-Aldrich). All assays were performed 48 hours after transfection.

Cell surface expression of HA-epitope tagged receptors was assayed using flow cytometry. In brief, cells were labelled with a monoclonal anti-HA antibody conjugated with APC (GG8-1F3.3.1; Miltenyi Biotec; dilution 1:50) with surface expression detected using a Sony ID7000 Spectral Cell Analyzer.

ADGRL4 versions (receptor-SmBiT) and mini-G protein or β-arrestin complementation were assayed using the NanoBiT PPI system (Promega). In brief, transfected cells were harvested using "Cell Dissociation Solution Non-enzymatic" (Sigma), cell counted and transferred in triplicate into a 384-well plate: 8 μL of cells per well were exposed to 2 μL NanoGlo substrate (1x NanoLuc Live cell substrate diluted in NanoGlo LCS Dilution buffer). After a 5 minute incubation on a shaker, luminescence was read using a Tecan Spark plate reader for 16 cycles. Ligands for positive controls were added at a fixed time-point during recording. Luminescence readout results were normalised by SmBiT and LgBiT construct expression levels with the following equation: [Coupling luminescence / (Total mini-G protein or β-arrestin construct expression luminescence * Total receptor construct expression luminescence)] * $1 \times 10^6$.

Positive control GPCRs with a C-terminal SmBiT and their respective ligands were used: 1) $G_q$: wild type HCRTR2 (ligand: 5 μM orexin-a peptide)[53]; 2) $G_s$: wild type turkey ADRB1 (ligand: 10 μM formoterol)[70]; 3) $G_{12}$: wild type GPR35 (ligand: 20 μM lodoxamide)[71]; 4) $G_o$: wild type APLNR (ligand: 20 μM CMF-019)[72].

Whole cell expression levels measurements of receptor-SmBiT and LgBiT-mini-G protein or LgBiT-β-arrestin were carried out using a similar method to the NanoBit system described above, but with the Nano Glo HiBiT lytic buffer. To measure the expression levels of receptor-SmBiT, purified LgBiT was added, and samples were incubated for 30 minutes before measuring luminescence. To measure the expression levels of LgBiT-mini-G protein or LgBiT-β-arrestin, the HiBiT-control protein was used and samples were incubated for 10 minutes before reading luminescence.

Additional C-terminal SmBiT tagged constructs without an N-terminal tag were generated: ΔHA-FL1; ΔHA-FLmut1; and ΔHA-CTF. Construct sequence fidelity was assessed using Sanger sequencing. G protein coupling assays were performed as described earlier with interactions between mutagenesis constructs (SmBiT tagged) and LgBiT-mini-$G_q$ (or LgBiT-mini-$G_o$) or LgBiT-mini-$G_q\Delta\alpha5$ (or LgBiT-mini-$G_o\Delta\alpha5$) constructs determined by luminescence and normalised as previously described.

### Construct design for structural studies
Four ADGRL4 constructs were designed and cloned into a pcDNA 3.1 expression vector for cryo-EM studies. Each construct contained N-terminal modifications to streamline detection, purification and tag removal: 1) a HA epitope tag (immunodetection); 2) a twin strep tag

(Strep-Tactin affinity purification); 3) His-10 tag (immobilised metal ion affinity chromatography (IMAC) purification); 4) HRV 3 C protease site (cleavage); 5) enhanced GFP (fluorescent detection); 6) TEV protease site (additional cleavage flexibility). The inclusion of both HRV 3 C and TEV sites permitted selective removal of the fused tags under different experimental conditions. At the C-terminus, each construct included a FLAG epitope tag flanked by 6 amino acid GS linkers. The four ADGRL4 structural studies constructs are described in detail in the results section. Additionally, ADGRL4-CTF2A contained a C-terminal tethered mini-$G_q$ (version A) that is unable to bind βγ due to a truncated N-terminal helix, whilst ADGRL4-CTF2B contained a C-terminal tethered mini-$G_q$ (version B) that does bind βγ due its N-terminal helix remaining intact. Mini-$G_q$ (version A) corresponded to the mini-$G_q$ used for coupling experiments[49]. Construct sequence fidelity was assessed using Sanger sequencing.

## Construct expression for structural studies
All four constructs for cryo-EM study were stably expressed in mycoplasma-free HEK293 GnTi⁻ TetR cell lines[73] (original HEK293 line authenticated by ATCC (CRL-1573). This cell line contains a selective genomic knockout of *N-acetylglucosaminyltransferase I* (*GnTI*; required for generating complex N-glycans) as well as a stable tetracycline-repressor plasmid (pcDNA6/TR) under blasticidin selection, allowing for tetracycline-inducible expression. As ADGRL4 has multiple N-terminal N-glycosylation sites, it was decided to stably express ADGRL4 constructs in a cell line without *GnTI* in order to avoid generating ADGRL4 with complex N-glycans.

In brief, ADGRL4 construct lentiviruses were assembled in Lenti-X 293 T cells (Takara) with titres assessed using the Lenti-X GoStix Plus kit (Takara). Low passage adherent HEK293 GnTi⁻ TetR cell lines were transduced and grown in DMEM (Gibco) supplemented by 5 μg/mL blasticidin (Melford) and 10% tetracycline-free fetal bovine serum (FBS) (BioSera), and then underwent fluorescence-activated cell sorting (FACS) for low-level leaky GFP expression using Becton Dickinson FAC-SAria Fusion or Invitrogen Bigfoot cell sorters. Sorted cells were then grown and converted to suspension culture in Freestyle medium (Gibco) supplemented by 5 μg/mL blasticidin and 5% tetracycline-free FBS. Once at scale, ADGRL4 construct expression was induced by exposure to 1 μg/mL tetracycline (Sigma) for 24-48 hours. At harvesting, whole-cell ADGRL4 construct expression levels were assessed using western blotting (HA epitope tag immunodetection) (NEB) and fluorescent microscopy (EVOS FL Fluorescence microscope). Cell surface expression levels were assessed using flow cytometry (HA epitope tag immunodetection (Miltenyi) using a Sony ID7000 Spectral Cell Analyzer). Harvested cells were counted and then snap-frozen in liquid nitrogen.

## Tetracycline time-course and small-scale membrane solubilisation
All four cryo-EM ADGRL4 constructs stably expressed in HEK293 GnTi⁻ TetR cells underwent tetracycline induction time-courses to assess optimal induction time. This was initially performed on adherent stable HEK293 GnTi⁻ TetR cells, and then performed again once the lines had been converted to suspension culture. In brief, stable lines were exposed to 1 μg/mL tetracycline (Sigma) for 24, 48 or 72 hours. At each time-point, cells were harvested, counted and assessed for construct cell-surface expression using flow cytometry (HA epitope tag immunodetection). Samples were then snap-frozen and later underwent small-scale membrane solubilisation to determine the optimal detergent conditions for each receptor as well as western blotting.

Cells underwent membrane solubilisation at 4 °C with either 1% DDM-0.1% CHS or with 1% LMNG-0.1% CHS. For DDM-CHS, cells were solubilised on a rotator for 1 hour, in contrast to 3 hours for LMNG-CHS. Solubilisation buffer for both DDM-CHS and LMNG-CHS contained 20 mM HEPES pH 7.5, 100 mM NaCl, 20% glycerol, 10 mM MgCl₂, 25 U/mL benzonase (Millipore), 100 μM TCEP, 25 mUnits/mL

apyrase (Sigma), 2 mM PMSF (ThermoFisher) and protease inhibitor (Roche). Detergent solubilised cells were then processed by ultra-centrifugation at 150,000 x*g* for 70 minutes using a TLA-55 rotor (Beckman) followed by supernatant collection. Solubilisates then underwent fluorescence-detection size-exclusion chromatography (FSEC) using a Shimadzu HPLC equipped with an Agilent Bio SEC-5 guard (5 μm, 300 Å, 7.8x50mm) and column (5 μm, 300 Å, 4.6x300mm) and a fluorescence detector.

## Large scale CTF2B purification
Following 48 hours of tetracycline induction, 2 L of 2.2×10⁶ cells/mL ADGRL4-CTF2B expressing HEK293 GnTi⁻ TetR cells were harvested and snap-frozen in liquid nitrogen. Cells were then thawed and solubilised at 4 °C on a rotator for 3 hours in 1% LMNG-0.1% CHS, 20 mM HEPES pH 7.5, 100 mM NaCl, 20% glycerol, 10 mM MgCl₂, 25 U/mL benzonase (Millipore), 100 μM TCEP, 25 mUnits/mL apyrase (Sigma), 2 mM PMSF (ThermoFisher) and protease inhibitor (Roche). Detergent solubilised cells were then ultracentrifuged at 136,000 x*g* for 70 minutes using a Type-45 Ti rotor (Beckman) followed by supernatant collection. The solubilisate was then loaded onto a Strep-Tactin XT 4Flow High Capacity Resin (IBA Lifesciences) by gravity flow. The resin was washed 5 times with 0.02% LMNG, 0.002% CHS, 20 mM HEPES pH 7.5, 100 mM NaCl, 20% glycerol, 10 mM MgCl₂, 100 μM TCEP, 2 mM PMSF (ThermoFisher) and protease inhibitor (Roche). Bound receptor was eluted by incubating the resin at 4 °C over a time-course with volumes of 50 mM biotin, 0.02% LMNG, 0.002% CHS, 20 mM HEPES, 100 mM NaCl, 20% glycerol, 10 mM MgCl₂, 100 μM TCEP, 2 mM PMSF (ThermoFisher) and protease inhibitor (Roche). Eluted protein was then concentrated using 100 kDa centrifugal filters (Amicon) and mixed with purified βγ at a 1:1.2 molar ratio on ice for 1 hour. The engineered protease sites were not cleaved off and the entire purified recombinant protein (including all purification and detection tags) was processed. Samples then underwent size exclusion chromatography (SEC) using a Shimadzu HPLC equipped with an Agilent Bio SEC-5 guard (5 μm, 300 Å, 7.8×50 mm) and column (5 μm, 300 Å, 4.6×300 mm) and a 280 nm detector. Running buffer comprised 0.03% LMNG, 20 mM HEPES pH 7.5, 150 mM NaCl, 10% glycerol, 10 mM MgCl₂, and 100 μM TCEP. Fractions were collected for SDS-PAGE gel electrophoresis and Coomassie staining. SEC fractions corresponding to the appropriate ADGRL4-CTF2B-βγ complex peak underwent further concentration before being applied to cryo-EM grids. At each step in the purification process, samples were collected for SDS-PAGE analysis and for FSEC and 280 nm detection using a Shimadzu HPLC.

## Cryo-EM grid preparation and data collection
For the preparation of cryo-EM grids, 3 μL of purified CTF2B/βγ complex was applied onto glow-discharged (2 min in Ar-Oxy 9-1 plasma chamber (Fischione), Forward Power of 38 W, Reflected Power of 2 W) 300 mesh Au grids (UltrAuFoil R1.2/1.3). The excess sample was removed by blotting for 3 s before plunge-freezing in liquid ethane (cooled to −181 °C) using a FEI Vitrobot Mark IV maintained at 100% relative humidity and 4 °C.

Cryo-EM data collection was performed at the Electron Microscopy Facility of the MRC Laboratory of Molecular Biology, Cambridge, UK. Two Datasets (4851 and 22950 movies) were automatically collected in two sessions using the EPU software (ThermoFisher) on the FEI Titan Krios microscope at 300 kV equipped with a Falcon 4i detector in counting mode. Both datasets were collected with a fluence of 60 $e^-$/Å² at 120,000x magnification (0.67 Å/pixel). The gain reference file was provided by the facility and used unmodified.

## Cryo-EM data processing
Raw movies in two datasets were imported into CryoSPARC v4.5.3[74] and the processing was performed there unless specified otherwise (Supplementary Fig. 4c). Overall, drift, beam-induced motion and dose

weighting were corrected with Patch Motion Corr. Contrast Transfer Function (CTF) fitting and phase shift estimation were performed using Patch CTF estimation. The exposures were manually curated: the only images kept had an estimated CTF resolution of <5 Å. This yielded two stacks of 4423 and 19395 movies. The two movie stacks were combined for further processing.

A total of 17,860,517 particles were autopicked using Blob Picker. Particle picks were curated to remove obvious junk picks by extraction with a box size of 360 pixels and then downsampled four times. The particles were subjected to two rounds of 2D classification and the clean particle stack (2,322,022 particles) was selected. These sorted particles were applied to ab initio reconstruction and classified into six classes. Iterative hetero refinement was performed with the six classes as an input. 374,008 particles were chosen from the good class and subjected to 2D classification. 339,728 particles belonging to the clean 2D classes were re-extracted with a box size of 400 pixels and processed for non-uniform refinement. Subsequently, reference-based motion correction and non-uniform refinement were performed, yielding a high-resolution density map with a global resolution of 3.14 Å. The resolution was determined using the gold-standard Fourier shell correlation using the 0.143 criterion.

### Model building and refinement
The initial ADGRL4-CTF2B model was obtained according to AlphaFold[75], and the $G_q$ heterotrimer was generated using ModelAngelo[76]. After docking the models into the corresponding EM density maps using UCSF Chimera[77], iterative rounds of manual adjustment and real-space refinement were performed in Coot[78] and real-space refinement in Phenix[79], respectively. MolProbity[80] was used to validate the final models and the comprehensive refining statistics are provided in Supplementary Table 2.

### Mass spectrometry
SDS-PAGE gel bands requiring identification were cut out and processed for mass spectrometry. In brief, samples underwent in-gel digestion followed by nano-scale capillary LC-MS/MS using an Ultimate U3000 HPLC (ThermoFisher). Peptides were analysed using a Orbitrap Q Exactive hybrid linear quadrupole ion trap mass spectrometer (ThermoFisher). Raw data was processed using Proteome Discoverer v3.0 (ThermoFisher).

### Tethered agonist region mutagenesis
Scanning alanine mutagenesis was performed on eight amino acid residues within the tethered agonist region. Based on our cryo-EM structure, only residues forming contacts with the receptor's orthosteric binding pocket within ≤3.9 Å were selected for mutagenesis (GAIN generic residue number scheme[56] in superscript): T407A[B.GPS.+1], H408A[B.S14.46], F409A[B.S14.47], Ile411A[B.S14.49], L412A[B.S14.50], M413A[B.S14.51], S414A[B.S14.52], S415A[B.S14.53]. Individual selected alanine mutations were cloned into versions of CTF1 (SmBiT tagged). Construct sequence fidelity was assessed using Sanger sequencing. G protein coupling assays were performed as described earlier with interactions between mutagenesis constructs (SmBiT tagged) and LgBiT-mini-$G_q$ or LgBiT-mini-$G_q\Delta\alpha5$ constructs determined by luminescence and normalised as previously described.

### Mutagenesis of putative 7TM activation motif regions
Alignment of 31 human protein-coding adhesion GPCRs was performed using GPCRdb[55]. Protein-coding status was defined as per the HUGO Gene Nomenclature Committee (HGNC)[81], Ensembl[82], and GeneCards[83] databases. ADGRF2 and ADGRE4P were not included as both human genes are considered to be pseudogenes. Scanning alanine mutagenesis was performed on eleven ADGRL4 amino acid residues within three putative 7TM activation motifs. Putative upper quaternary core residues: F505[3.44], M508[3.47], W631[6.53] (Q656[7.49] also included as it forms a hydrogen

bond with W631[6.53] in the canonical upper quaternary core motif); putative P/F/W/LφφG motif residues: F625[6.47], L626[6.48], L627[6.49], G628[6.50]; putative H(N)L(M)Y motif residues: H514[3.53], L515[3.54] and Y516[3.55]. Individual selected alanine mutations were cloned into versions of CTF1 (SmBiT tagged). Construct sequence fidelity was assessed using Sanger sequencing. G protein coupling assays were performed as described earlier with interactions between mutagenesis constructs (SmBiT tagged) and LgBiT-mini-$G_q$ or LgBiT-mini-$G_q\Delta\alpha5$ constructs determined by luminescence and normalised as previously described.

### Reporting summary
Further information on research design is available in the Nature Portfolio Reporting Summary linked to this article.

## Data availability
The source data underlying Fig. 1d-h, Fig. 3e, Supplementary Fig. 1, Supplementary Fig. 2 and Supplementary Fig. 5 are provided as a Source Data file. Cryo-EM maps have been deposited in the Electron Microscopy Data Bank (EMDB) under accession code EMD-53437. The atomic coordinates have been deposited in the Protein Data Bank (PDB) under accession code PDB 9QXL. Source data are provided with this paper.

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

## Acknowledgements

This work was supported by funding from Cancer Research UK (CRUK) (grant holder D.M.F.; grant number: RCCPOB-May22\100008), from the Medical Research Council, and from the Department of Oncology, University of Cambridge (fellowship holder: D.M.F.). C.G.T. was sup-ported by core funding from the Medical Research Council (MRC U105197215). Mass spectrometry analysis was conducted at the Biolo-gical Mass Spectrometry and Proteomics Facility of the MRC Laboratory of Molecular Biology (LMB), Cambridge, UK. We thank Dr. Catarina Franco and Dr. Farida Begum for their assistance with mass spectro-metry. Cell sorting was performed at the Flow Cytometry Facility of the MRC LMB, and we are grateful to Dr. Pier Andrée Penttilä, Dr. Fan Zhang, and Yichen Li for their support. We acknowledge the LMB Electron Microscopy Facility for access and support of electron microscopy sample preparation and data collection, and the LMB scientific com-puting for technical support. For the purpose of open access, the MRC Laboratory of Molecular Biology has applied a CC BY public copyright licence to any Author Accepted Manuscript version arising.

## Author contributions

D.M.F. designed, planned and managed the overall project. D.M.F. developed and performed G protein coupling experiments (designed and cloned coupling constructs, performed and interpreted assays), developed and performed the mammalian expression strategy for structural studies (designed and cloned constructs for structural stu-dies, developed stable cell lines), and developed and performed the purification strategy. A.D. and J.C.P. provided guidance to the coupling experiments. C.G.T. provided guidance to the structural studies experiments. P.C.E. assisted with cell line expression. Q.C., A.G., and D.M.F. prepared cryo-EM grids and performed cryo-EM data collection. QC performed model building. Q.C., D.M.F. and C.G.T. carried out structure analysis. D.M.F wrote the manuscript, with all authors contributing.

## Competing interests

The authors declare the following competing interests: AD and JCP are employees and shareholders of Nxera Pharma; CGT is a shareholder, consultant and member of the Scientific Advisory Board of Nxera Pharma. Unique materials described in this paper are freely available upon reasonable request. The remaining authors declare no competing interests.
