## [Transparent Peer Review file · Nature Communications]

Structure of the G_q-coupled adhesion receptor ADGRL4

Corresponding Author: Dr David Favara

Version 0:

Reviewer comments:

Reviewer #1

(Remarks to the Author)

The current manuscript by Chen et al. documents a remarkable and long sought after achievement with the reporting of the metabotropic coupling of the aGPCR ADGRL4 to a G_q pathway, and the documentation of the cryo-EM structure of the active form of its CTF subunit at 3.1 Å resolution. The results are exceptionally clear, well controlled and add significantly to the rapidly enlarging knowledge on the structure-function duality of aGPCR. The role of L4 in cancer biology underlines the importance of this manuscript, as indeed the structure may lend themselves to expedited development and testing of compounds hedging in the activity of L4 in cancers.

There are several minor and one major comment, concerning the signaling network within the CTF, that I would like to ask the authors to tend to. Once they have solidified the significance of the properties of their newly identified L4 CTF structure I have no objections to seeing this fine manuscript published in Nature Communications.

General assessment of the manuscript:

What are the noteworthy results? YES

Will the work be of significance to the field and related fields? YES

How does it compare to the established literature? DESCRIBED BY THE AUTHORS

Does the work support the conclusions and claims? MOSTLY, SEE COMMENTS or is additional evidence needed? SOME, SEE MY SUGGESTIONS

Are there any flaws in the data analysis, interpretation and conclusions? NO

Is the methodology sound? YES

Does the work meet the expected standards in your field? YES

Is there enough detail provided in the methods for the work to be reproduced? YES

MAJOR POINTS

Based on the novel findings of the authors that L4 is indeed found to couple to a G protein, i.e. G_q, extending if not negating the team's previous claims on L4 signalling, it would be prudent to consolidate this finding by additional investigations into the Stachel-binding pocket relationship of the L4-CTF1 protein similar to previous studies that described the alpha-helical structure of the Stachel once liberated from the GAIN domain and bound to its orthosteric binding pocket in the 7TM domain. The authors themselves compare the Stachel-7TM contacts in lines 189-191 but do not test the significance of these observations on L4 signalling, which is an obvious experiment logically following from their discoveries.

Therefore, the authors should capitalise on their newly identified contacts of the Stachel and the binding pocket as reported in Fig. 3b,c, mutagenise the contacting residues on the respective Stachel or 7TM side of the interface, and test mutant L4-NTF1 proteins for their basal activity with their L4-CTF1-SmBit1/LgBit-G_q assay akin to similar binding relationship studies reported for Gpr133 (e.g. see Fig. 2h [Stachel residues mutagenesis] and I [pocket residue mutagenesis] in 10.1038/s41586-022-04619-y), Gpr56 (e.g. see Fig. 3c in 10.1038/s41586-022-04575-7), or Lphn3 (e.g. see Fig. 3d in 10.1038/s41586-022-04575-7).

Along the same line, the now available L4 signalling assay lends itself to clarify the presence or absence of previously identified activation motifs within the 7TM domain of other aGPCRs (i.e. the upper quaternary core, P/F/W/LφφG, H(N)L(M)Y, reviewed in 10.1016/j.tibs.2023.05.007). The authors are asked first determine whether any of the previously characterised

signaling motifs are present in the 7TM domain of L4. If so, they should mutagenise them and test them for changes in Gq coupling.

MINOR POINTS

Abstract + Introduction

“... are a 33-member subfamily of class B ...”. They are not a subfamily but a family. And in humans there are only 32, not 32 homologs (ADGRE4/EMR4 is a pseudogen in humans). As the authors refer to the human family, please correct also the number wherever it appears in the manuscript.

38: when citing ref. 3 as initial discovery of the Stachel mechanism, also cite ref. 10.1073/pnas.1421785112

38: the receptor-integral Stachel is not a peptide, but part of the contiguous full-length or at least CTF receptor portion. Only when the Stachel is generated by peptide synthesis, e.g. to be added in pharmacological assays, the “Stachel” should be called a “Stachel peptide”

42: There is more suitable reference to an original research paper instead of the review in ref. 6:
10.1016/j.molcel.2020.12.042

45-47: The cited ref. 16 does not contain a 7TM but an ECR structure and is thus unsuitable to validate the authors' statement: “This activation model is supported by recently solved cryo-EM structures of active-state aGPCRs”

52-54: “In general, higher ADGRL4 expression within the tumour microenvironment across the majority of these tumours correlates with more aggressive disease in vitro and in vivo”. Cite reference(s) supporting this claim.

90 ff.

“Mutated Full-Length 1' (FLmut1) was the full-length construct containing the alanine substitutions H405A and T407A at the GPCR proteolysis site (GPS), which prevented 92 autocatalytic cleavage and hence NTF and CTF dissociation”.

- Ext. Data Fig. 3c shows a WB of all these mutants but not the L4 protein with intact GPS to allow judgement if the WT receptor is autoproteolysed to start with. Include a full length WT protein too and run it along with the mutants.

- It is puzzling that the human FL WT L4 protein should be ca. 78 kDa but the GPS-deficient versions shown by the authors (FLmut2, FLmut2.1) run at double the size (see Ext. Fig. 3c). Elaborate on this conflict between prediction and data.

- Similarly: L4-CTF (CTF2) should be 32 kDa but runs at appx. 60 (see Ext. Fig. 3c). Elaborate on this conflict between prediction and data.

- Please indicate the expected sizes in kDa for all proteins (FL, CTF, CTF-SmBit fusions) tested in WB so the reader can judge the depicted bands.

Results

Line 176 ff.

Despite the L4-CTF 7TM structure at hand, the authors have forgone the chance to clearly comment on the exceptional placement of L4 within the aGPCR family. Based on its adhesive domains in the ECR it was classically considered to be an ADGRE member, but the primary sequence of the 7TM domain placed it firmly within the ADGRL subfamily (e.g. in 10.1016/j.ygeno.2003.12.004, 10.1124/pr.114.009647, 10.1111/nyas.14192).

With the structure of the active 7TM in hand now, and the existence of atomistic structures of other L and E 7TM domains to compare it to (L3 and E5), the authors should comment and evaluate the kinship of L4 to the L and E subfamilies, generate superimposition images similar to their Fig. 4. The integration of F1 in the comparisons there muddies the clarity of the data presentation.

Could the authors make a clear statement, perhaps in the heading of the section explaining the results in Fig. 4, to render their verdict on the similarity and placement of L4 between the E and L subfamily clear?

Fig. 1

Fig. 1d-h: although the caption states that three biological replicates were obtained each containing 3 technical replicates (i.e. 9 data points), these 9 data points are only plotted in Fig. 1d. In Fig. e-h only three datapoints are visible. What do they correspond to and why does the data presentation differ from the one in Fig. 1d?

Generally, the authors should add their statistical result comparing FL1 to CTF1 and FLmut1 to CTF1 coupling to the intact

G protein (just as annotated in Fig. 1d) in Fig. 1e-g even though the results indicated no significant difference.

Ext. Data Fig. 1

Ext. Data Fig. 1a: although the CTF1 variant was N-terminally tagged with an HA tag it did not show at the cell surface in neither combination with or without an intracellular pathway component. This observation has been made for other aGPCRs before, while others reported that CTF receptor variants can traffick to the plasma membrane and can get detected there. Please describe this result and comment on your conclusions in the text where you refer to the data, i.e. in the block 108-111. Could the coupling of L4-CTF1 to Gq be greater than of L4-FL because the subcellular location of the receptor is different? Could the receptor be internalised or rapidly get internalised because the NTF is missing? This is a particular possibility that the authors considered for their L4-CTF2 variant later introduced and described in the manuscript, where they write in main text: "ADGRL4-CTF2 had the lowest with close to no surface expression suggesting either instability, misfolding and/or rapid turnover of the potentially constitutively active receptor" (lines 142-144). So, if this applies to the L4-CTF1 variant too, do the authors assume that their Gq coupling to L4-CTF protein occurs in the ER or en route to the plasma membrane, as opposed to at the cell surface? Is this reasonable?

Reviewer #2

(Remarks to the Author)

The current study provides the structure of the C-terminal fragment of the adhesion GPCR, ELTD-1. ELTD-1 has roles in cancer and angiogenesis, so understanding the structural basis of its activation mechanism and G protein coupling is significant. Overall the structural work is of high quality and demonstrates that ELTD-1 is activated by its tethered agonist. The pose of the tethered agonist within the 7TM orthosteric resembles the poses of tethered agonists for other adhesion GPCRs with recently solved structures. The structural and signaling data provide a potential explanation of why ELTD-1 has been considered a poor G protein coupler or lacked this ability altogether. The ELTD-1 structure was of a complex with mini-Gq heterotrimer. The position of the Gαq alpha-5 helix that engaged the 7TM was in a pose that suggested weaker Gq coupling and did not resemble other active-state GPCR structures in which this helix is inserted more deeply into the intracellular opening of the 7TM. There are some potential caveats of the signaling data that require additional experimentation or explanation. Most concerning is the use of N-terminal HA tag ELTD-1 expression constructs. For evaluation of signaling from the full length wildtype and cleavage defective mutants, this tag is probably not an issue, but for evaluation of the CTF constructs it certainly is. The original study and many subsequent studies that describe the adhesion GPCR tethered agonist mechanism demonstrate clearly that when any sequence other than an initiator methionine is appended onto the N-terminus of adhesion GPCR CTFs, that this abrogates tethered agonism. The series of papers in 2022 from multiple labs that described several adhesion GPCR tethered agonist-activated structures capitalized on this finding and utilized the "naked" methionine CTF constructs. It was presumed with good assumption that the methionine was cleaved off by methionyl aminopeptidase as is the case for most cellular proteins and that no density for a N-terminal methionine was observed in the structures. So, if an HA-tagged CTF construct was used throughout the signaling studies of Figure 1, this requires additional examination with a non-tagged construct. One additional point that the authors do touch on is the odd results about positive G_o coupling that is statistically significant, but is brushed off because the coupling appeared stronger to the full length receptor and this does not fit the model. The statistically significant Gq coupling to the CTF construct did fit the model, so this was accentuated in the write-up. Is there any assay problem with mini-G_o? Mini-G_o in a 2021 GPR97 structural study was modified with a non-native palmitate on its alpha 5 helix, providing a completely artifactual mode of receptor / G protein engagement.

For the structural studies, it is not completely clear whether the issue with modified N-termini pertains. Four different expression constructs for use in receptor preparation for structural determination are described, and it might be the case that cleavage of the engineered protease sites produced an authentic N-terminus. This should be better clarified in the manuscript. How did the structure(s), which begin with the threonine of the authentic tethered agonist arise?

Reviewer #3

(Remarks to the Author)

This study by Favara and colleagues presents a comprehensive investigation into the signaling and structural properties of ADGRL4, an orphan adhesion GPCR that has been implicated in a variety of cancers. Despite its clinical significance, ADGRL4 has remained poorly understood, and previous studies failed to detect its coupling to heterotrimeric G proteins, leading to uncertainty about its signaling mechanisms. In this study, the authors use a highly sensitive NanoBIT-based bioluminescence assay to demonstrate that ADGRL4 exhibits weak but specific coupling to the Gq protein, while no robust interaction is observed with other G protein subtypes (Gs, G12, Go) or with β-arrestin 1/2. This weak coupling, which was previously undetectable using standard approaches, is further supported by cryo-EM structural analysis. The authors then determined the structure of ADGRL4 in complex with mini-Gq at an overall resolution of 3.1 Å. The structure reveals that ADGRL4 maintains the canonical architecture of other aGPCRs and is activated by its endogenous tethered agonist, which adopts a short α-helical conformation and binds within the orthosteric pocket. Notably, the interface between ADGRL4 and Gq exhibits markedly fewer and weaker contacts compared to other structurally characterized Gq-coupled aGPCRs, providing a structural rationale for the observed weak signaling output.

1. A major concern is whether the Gq-coupling activity of ADGRL4 demonstrated in this study accurately reflects its

physiological signaling. The mini-G protein constructs used are highly engineered, and the signaling assays rely on overexpression of both the receptor and G proteins in heterologous systems. It remains unclear whether ADGRL4 can activate endogenous Gq signaling in native cells. As the authors mentioned, the JAK/STAT3 and MAPK/ERK pathways can be driven by various upstream signaling molecules. To address this, it is important to assess whether the CTF1 construct can stimulate wild-type Gq signaling by measuring calcium mobilization or IP3 accumulation. Ideally, these experiments should be performed in native cells without overexpression of G proteins. However, even studies in heterologous systems with co-expression of CTF1 and wild-type Gq would provide critical supporting evidence.

2. CTF1 is supposed to be a constitutively active GPCR. However, in Fig. 1h, why CTF1 is less active in β -arrestin recruitment compared to FL1 and FLmut1?

3. There is ongoing debate in the adhesion GPCR field regarding whether all aGPCRs, or how many, are activated by Stachel peptides and whether cleavage of the NTF necessarily leads to its dissociation from the CTF. In some cases, Stachel peptides may not serve as activators, or they may represent only one of multiple activation mechanisms under physiological conditions. Although the cryo-EM structure clearly shows that the Stachel peptide binds to ADGRL5 in a mode similar to that observed in other aGPCR structures, this does not necessarily establish it as the endogenous agonist. For instance, there are structures of opioid receptor-G protein signaling complexes bound to an inverse agonist (<https://doi.org/10.1038/s41589-024-01812-0>), suggesting that structural engagement alone does not confirm physiological agonism. It is therefore recommended to test the free form of the Stachel peptide in the NanoBiT-based G protein recruitment assays. Have any studies examined the function of the free Stachel peptide, e.g. whether it can stimulate the JAK/STAT3 or MAPK/ERK pathways via ADGRL4?

4. As the authors noted, several structures of adhesion GPCRs bound to their tethered peptides have been reported, all of which exhibit a similar ligand-binding mode. In fact, AlphaFold3 can predict an active-state structure of ADGRL4 bound to its tethered agonist peptide, which closely resembles the experimentally determined structure reported in this study (RMSD < 1 Å for C α atoms and ~2 Å for all atoms). While this does not diminish the value of the experimentally resolved structure, especially considering that it provides more information on receptor signaling, it would be informative to include a direct comparison with the AlphaFold3 model and discuss the implications of AI-based structure prediction for future drug development targeting ADGRL5. It may also be worth discussing whether AI tools can accurately predict an inactive conformation of ADGRL5.

5. To enable reproducibility, it is recommended to include more detailed information on the design of the signaling constructs, specifically, the exact residue ranges of ADGRL5 used in each construct. For people outside the adhesion GPCR field, the start of the Stachel sequence may not be readily apparent, so clarifying this would improve accessibility and help other researchers to reproduce the data.

6. It is recommended to revise the title. Many GPCRs are implicated in cancer. This paper does not explore the functional roles of ADGRL4 in cancer. And it is not a well established cancer target.

Version 1:

Reviewer comments:

Reviewer #1

(Remarks to the Author)

This is a revised version of a manuscript reporting the G protein bound structure of L4-CTF and its function as a Gq coupled receptor. Most of my comments were addressed one regarding the activation motifs was not.

MAJOR POINTS

ORIGINAL REQUEST

Therefore, the authors should capitalise on their newly identified contacts of the Stachel and the binding pocket as reported in Fig. 3b,c, mutagenise the contacting residues on the respective Stachel or 7TM side of the interface, and test mutant L4-NTF1 proteins for their basal activity with their L4-CTF1-SmBit1/LgBit-Gq assay akin to similar binding relationship studies reported for Gpr133 (e.g. see Fig. 2h [Stachel residues mutagenesis] and I [pocket residue mutagenesis] in 10.1038/s41586-022-04619-y), Gpr56 (e.g. see Fig. 3c in 10.1038/s41586-022-04575-7), or Lphn3 (e.g. see Fig. 3d in 10.1038/s41586-022-04575-7).

COMMENT

I commend the authors on the results obtained with agonist mutagenesis confirming the relevance of the active L4 structure.

ORIGINAL REQUEST

The now available L4 signalling assay lends itself to clarify the presence or absence of previously identified activation motifs within the 7TM domain of other aGPCRs (i.e. the upper quaternary core, P/F/W/L ϕ ϕ G, H(N)L(M)Y, reviewed in 10.1016/j.tibs.2023.05.007). The authors are asked first determine whether any of the previously characterised signaling motifs are present in the 7TM domain of L4. If so, they should mutagenise them and test them for changes in Gq coupling.

COMMENT

The authors looked for the presence of the activation motifs in L4, discovered some, but did not test their functional relevance with their signalling assay. Instead they referred that to a future manuscript (as for several other requests made by all three referees).

While I understand the time and resources required for these assays to run I would like to emphasise the need to connect structural insights to functional data, in particular as the few manuscripts that have done that in the past are still rare. In order for the manuscript, to be published in Nature Communications, to add to the general knowledge on how adhesion GPCRs principally work, the current principles such as the found activation motifs need to be tested experimentally. Currently, the manuscript fails to do this and only provides high-quality details about an individual receptor of this family. I would thus ask the authors again to supply signalling tests of L4 variants containing mutagenised UQC, FLLG and HLY motifs, when they have ascertained that the mutated CTF proteins are expressed similarly to the WT L4-CTF. Also: when the authors refer to the activation motifs in the text they should not cite Seufert et al., 2023, which is a review of the state-of-the-art at the time containing all the references to primary literature. I pointed this paper out to the authors to make it easier for them to get an overview of the current state of knowledge on such motifs. But the authors should of course rather cite the original work that discovered the motifs.

MINOR POINTS

ORIGINAL REQUEST

Fig. 1

Generally, the authors should add their statistical result comparing FL1 to CTF1 and FLmut1 to CTF1 coupling to the intact G protein (just as annotated in Fig. 1d) in Fig. 1e-g even though the results indicated no significant difference.

COMMENT

There is still one significance marker missing in Fig. 1d, i.e. between FLmut1 (grey) vs. CTF1 (grey). Please add this to the figure.

All other comments were satisfactorily addressed.

Reviewer #2

(Remarks to the Author)

The authors have largely brushed off my concerns. They provided partially incomplete explanations of revised text as an attempt to clarify issues raised by the critiques. Regarding the appendage of N-terminal tags to CTF-only versions of the adhesion GPCRs, the authors chose to cherry-pick specific references that are attributed to one research lab, and to ignore many other studies that were perhaps performed more rigorously, to support their position that N-terminal tags do not impact ELTD-1 tethered agonism and signaling. An issue with each of the referenced studies that parallels potential issues with the ELTD-1 constructs used in the current study is that there is no calibration of the signaling assays that directly compared whether there might be much stronger signaling of naked N-terminal CTF constructs versus the N-terminal tagged versions. Could the signaling assays with mini-G proteins even support measuring a range of low efficacy activation to very high efficacy versions? It might be best to perform the signaling measurements using an orthogonal assay platform and to compare to the mini-G protein results. These issues are especially pertinent to the current study because it claims to show weak Gq coupling and no Go coupling (although there appear to be statistically significant differences amongst the updated Go assays). What if the reason the signaling appears to be weak is because there is an abnormal N-terminal tag blocking full access of the tethered agonist to the orthosteric site? The authors do see a pose of the tethered agonist within the orthosteric site that is reminiscent of the poses of tethered agonists in other active state adhesion GPCRs, but again it is concerning that the high resolution density of the tethered agonist happened to stop precisely at the first authentic residue of the tethered agonist and there was no density or resolved structure for the artificial tag residues that remained immediately N-terminal to this portion of the receptor. Hand waving disclaimer statements that this region was just unordered should not make this major issue just go away. Were the tags spontaneously cleaved off right before the threonine residue? Could this be why the constructs signal and no density for the tags appeared? The authors state that the major goal of the research was to solve the structure of active-state ELTD1, but they also declare that the signaling "is solved" for ELTD-1. The former is clear, but the latter could be an overclaim based on the concerns noted about the N-terminal tags.

Reviewer #3

(Remarks to the Author)

The authors provide justified responses to my comments without providing any additional experimental data. I would recommend at least including discussion of the limitations of the structural insights and signaling assays using engineered G

proteins.

Regarding cancer involvement, as I mentioned, it is often possible to find references suggesting direct or indirect associations with cancer for most GPCRs. However, such associations alone do not necessarily justify considering a GPCR as a valid therapeutic target for cancer treatment. On the other hand, I am not strongly against the title since it is not a major issue.

Version 2:

Reviewer comments:

Reviewer #1

(Remarks to the Author)

This is the second revision of a manuscript reporting the G protein bound structure of L4-CTF and its function as a Gq coupled receptor. Most of my comments were now all satisfactorily addressed and I recommend acceptance of the paper.

MAJOR POINTS

ORIGINAL REQUEST

The now available L4 signalling assay lends itself to clarify the presence or absence of previously identified activation motifs within the 7TM domain of other aGPCRs (i.e. the upper quaternary core, P/F/W/LφφG, H(N)L(M)Y, reviewed in 10.1016/j.tibs.2023.05.007). The authors are asked first determine whether any of the previously characterised signaling motifs are present in the 7TM domain of L4. If so, they should mutagenise them and test them for changes in Gq coupling.

COMMENT: The authors have now fulfilled my initial request to analyse the role of the identified 7TM activation motifs and the tethered agonist sequence in full. I congratulate them for the meticulous analyses they have undertaken to bind their findings on the L4 structure to its activation by its tethered agonist. In particular to see that mutating the TA core sequence from the +2 position onwards profoundly compromises the activity of the receptor is reassuring and will support the function of L4 as a G protein coupled receptor, which was previously disputed. Also addressing the concern of the potentially harmful effect of the N-terminal HA-tag, that was raised by another reviewer, was important and underlines the findings in the coupling assays.

The results will be most useful for others to compare their molecular pharmacological and structural findings with and learn from the L4 receptor.

MINOR POINTS

ORIGINAL REQUEST

There is still one significance marker missing in Fig. 1d, i.e. between FLmut1 (grey) vs. CTF1 (grey). Please add this to the figure.

COMMENT: This has been resolved now.

Reviewer #2

(Remarks to the Author)

The authors have experimentally addressed the critiques and provided adequate explanations and interpretations of their data.

I think it is much more than nuance to demonstrate that the new tested constructs with authentic tethered agonists provide higher apparent efficacies than the HA-tagged constructs. This should have been examined previously during the peer review of the major papers that the current authors primarily used for the design of their original submission constructs. (First revision paraphrased response: "N-terminal tags did not affect coupling/signaling in references, X, Y, Z"). The tags do affect coupling/signaling/recruitment and it is good that the revised manuscript will help to rectify this for the field. The authors might consider moving this new data from Extended Data Figure (1) into the main manuscript figure set. It's not convenient to do this in a twice revised manuscript, but it would be best not to bury this in the ED.

The response to the absence of ordered data for the HA tag in the structure is hand waving. If there is no data, there is no data, but if the HA tagged tethered agonist binds to the orthosteric site in a less efficient manner than the authentic tethered agonist (not tested), it could be that the structure does not represent a fully-active 7TM state and therefore could explain the weakly engaged pose that Gq exhibited. I think the authors should at least acknowledge this possibility with a sentence in the discussion.

Otherwise, the manuscript is improved.

Reviewer #3

(Remarks to the Author)

The authors have provided additional mutagenesis data on the tethered agonist and key 7TM activation motifs of the receptor, which substantially strengthen their structural insights. The additional discussion addressing the limitations of the study is also satisfactory and resolves my earlier concerns. I have only one minor suggestion: please update the label of ADGRL4 to ADGRL4-CTF2B in the main Figure 2, and replace Gq with mini-Gq in the main Figure 6, along with updates to the figure legends for accuracy.

We thank the reviewers for their detailed assessment of our work and their recommendations
for improving the manuscript.

Below are our replies to all comments in blue (all line numbers referenced in our replies to the
reviewers refer to the manuscript with Track Changes visible). All additions to our manuscript
are visible as Tracked Changes.

Below is a summary of all major figure and table revisions to our manuscript:

9 A) **Fig.1d**: added additional replicates to the positive control for G_q to ensure n=3.

B) **Fig.1e-h**: added additional replicates added to ensure n=3 for all signalling
experiments.

C) Performed alanine mutagenesis and added new panel **Fig.3e** showing tethered
agonist alanine mutagenesis G_q complementation assay results.

D) **Extended Data Fig. 1a**: updated graph to show representative data of n=3.

E) Added new figure (**Extended Data Fig.5**) reporting expanded tethered agonist
alanine mutagenesis G_q complementation assays.

F) Updated **Extended Data Table 1** with new statistics after adding extra replicates to
ensure n=3 for all signalling experiments.

G) Added new table (**Extended Data Table 3**) reporting mutagenesis statistics.

H) Updated the "**source_data.xlsx**" file with new raw data as reported.

**Reviewer #1 (Remarks to the Author):**

The current manuscript by Chen et al. documents a remarkable and long sought after
achievement with the reporting of the metabotropic coupling of the aGPCR ADGRL4 to a G_q
pathway, and the documentation of the cryo-EM structure of the active form of its CTF
subunit at 3.1 Å resolution. The results are exceptionally clear, well controlled and add
significantly to the rapidly enlarging knowledge on the structure-function duality of aGPCR.
The role of L4 in cancer biology underlines the importance of this manuscript, as indeed the
structure may lend themselves to expedited development and testing of compounds hedging
in the activity of L4 in cancers.

There are several minor and one major comment, concerning the signaling network within
the CTF, that I would like to ask the authors to tend to. Once they have solidified the
significance of the properties of their newly identified L4 CTF structure I have no objections to
seeing this fine manuscript published in Nature Communications.

General assessment of the manuscript:
What are the noteworthy results? YES
Will the work be of significance to the field and related fields? YES
How does it compare to the established literature? DESCRIBED BY THE AUTHORS
Does the work support the conclusions and claims? MOSTLY, SEE COMMENTS or is
additional evidence needed? SOME, SEE MY SUGGESTIONS
Are there any flaws in the data analysis, interpretation and conclusions? NO
Is the methodology sound? YES
Does the work meet the expected standards in your field? YES
Is there enough detail provided in the methods for the work to be reproduced? YES

We thank the reviewer for their detailed assessment of our work and their recommendations
for improving the manuscript. We will reply to each point raised below in blue text. Line
numbers below refer to the version of the manuscript with Track Changes visible.

**MAJOR POINTS**

Based on the novel findings of the authors that L4 is indeed found to couple to a G protein,
i.e. Gq, extending if not negating the team's previous claims on L4 signalling, it would be
prudent to consolidate this finding by additional investigations into the Stachel-binding
pocket relationship of the L4-CTF1 protein similar to previous studies that described the
alpha-helical structure of the Stachel once liberated from the GAIN domain and bound to its
orthosteric binding pocket in the 7TM domain. The authors themselves compare the Stachel-
7TM contacts in lines 189-191 but do not test the significance of these observations on L4
signalling, which is an obvious experiment logically following from their discoveries.

Therefore, the authors should capitalise on their newly identified contacts of the Stachel and
the binding pocket as reported in Fig. 3b,c, mutagenise the contacting residues on the
respective Stachel or 7TM side of the interface, and test mutant L4-NTF1 proteins for their
basal activity with their L4-CTF1-SmBit1/LgBit-Gq assay akin to similar binding relationship
studies reported for Gpr133 (e.g. see Fig. 2h [Stachel residues mutagenesis] and I [pocket
residue mutagenesis] in 10.1038/s41586-022-04619-y), Gpr56 (e.g. see Fig. 3c in
10.1038/s41586-022-04575-7), or Lphn3 (e.g. see Fig. 3d in 10.1038/s41586-022-04575-7).

We thank the reviewer for this suggestion and have performed alanine mutagenesis
experiments on amino acid residues within ADGRL4's tethered agonist region which form
contacts \$\leq 3.9\$ Å in our cryo-EM model with the orthosteric binding pocket. As expected, results

show that seven residues within ADGRL4's tethered agonist region contribute to activation
(H408, F409, I411, L412, M413, S414, S415), the most critical being the hydrophobic M413
and F409 residues.

The following new figures and table have been added to the manuscript to report the above
results:

- • Fig. 3e,
- • Extended Data Fig. 5
- • Extended Data Table 3

The following text has been added to the manuscript to describe the above results.

Within the results section (line 216-230):

“To identify the side-chains critical for ADGRL4 activation, alanine scanning
mutagenesis was performed on eight tethered agonist residues (T407-S415) forming
contacts (≤ 3.9 Å) with residues within the orthosteric binding pocket, as determined by
our cryo-EM structure. Mutations were introduced into the constitutively active
ADGRL4-CTF1 signalling construct (C-terminally SmBiT-tagged) and assessed for G_q
coupling using the NanoLuc complementation assay. Seven of the eight substitutions
significantly reduced G_q signalling, with six (H408A, F409A, I411A, L412A, M413A,
S414A) reducing G_q activity by >50% of wild-type CTF1. M413A abolished activity
entirely ($p < 0.0001$) and F409A produced the second largest decrease to 12% of wild-
type CTF1 ($p < 0.0001$) (Fig. 3e, Extended Data Fig. 5, Extended Data Table 3). These
findings are consistent with the cryo-EM structure and identify that the most important
amino acids in the tethered agonist region for ADGRL4 activation are M413 followed
by F409 within the hydrophobic block F409-x-I411-L412-M413 (where x is residue
A410 which does not form any contacts in our cryo-EM model). “

Within the discussion section (line 291-296):

“Our alanine mutagenesis results show that seven residues within ADGRL4's tethered
agonist region contribute to activation (H408, F409, I411, L412, M413, S414, S415),
the most critical being the hydrophobic M413 and F409 residues. These results largely
conform to the canonical FX ϕ ϕ ϕ X ϕ motif noted in other aGPCRs (Demberg et al.,
2017, Ping et al., 2022, Xiao et al., 2022) (X=any residue; ϕ =a hydrophobic residue
such as Ile, Leu and Met), except that ADGRL4 contains a polar serine (S415) at the
final ϕ position.”

Within the methods section (line 568-576):

“Tethered agonist region mutagenesis

Scanning alanine mutagenesis was performed on eight amino acid residues within the tethered agonist region. Based on our cryo-EM structure, only residues forming contacts with the receptor’s orthosteric binding pocket within ≤ 3.9 Å were selected for mutagenesis: T407A, H408A, F409A, Ile411A, L412A, M413A, S414A, S415A. Individual selected alanine mutations were cloned into versions of CTF1 (SmBiT tagged). Construct sequence fidelity was assessed using Sanger sequencing. Signalling assays were performed as described earlier with interactions between mutagenesis constructs (SmBiT tagged) and LgBiT-mini-G_q or LgBiT-mini-G_qΔa5 constructs determined by luminescence and normalised as previously described.”

Along the same line, the now available L4 signalling assay lends itself to clarify the presence or absence of previously identified activation motifs within the 7TM domain of other aGPCRs (i.e. the upper quaternary core, P/F/W/LφφG, H(N)L(M)Y, reviewed in 10.1016/j.tibs.2023.05.007). The authors are asked first determine whether any of the previously characterised signaling motifs are present in the 7TM domain of L4. If so, they should mutagenise them and test them for changes in Gq coupling.

We thank the reviewer for this suggestion and have carefully examined the presence of the previously characterised activation motifs in the ADGRL4 7TM domain, as reviewed in Seufert et al (10.1016/j.tibs.2023.05.007) and provide the following structural insights:

- Upper Quaternary Core (UQC) Motif:
 - The UQC motif, defined as F3.44, M3.47, F5.43/F5.47, and W6.53, is proposed to stabilise the TM helix bundle in active conformations of ADGRG2 and ADGRG3.
 - In the ADGRL4 structure, three of the four canonical upper quaternary core (UQC) residues are conserved, including W6.53, which is stabilised by a side chain interaction with Q7.49. Unlike in ADGRG2, ADGRG4, and ADGRL3, this interaction does not involve a hydrogen bond, and W6.53 does not form direct contacts with the tethered agonist), suggesting a distinct local packing arrangement within the 7TM core.

○ The F5.43/F5.47 component of the motif is absent in ADGRL4, as it is in other
aGPCRs such as ADGRF1, ADGRL3, and ADGRG5, suggesting this position
may be dispensable or variable across aGPCR subfamilies.

• F ϕ ϕ G (FLLG) Motif:

○ The conserved FLLG motif is present in ADGRL4 and mediates the TM6 kink.
Importantly, Q7.49, which interacts with W6.53, is also closely packed against
this FLLG motif, implying an analogous structural role in stabilizing TM6
conformation during activation.

• H(N)L(M)Y Motif:

○ The H3.53-L3.54-Y3.55 motif is also conserved in ADGRL4. Among
these, L3.54 directly contacts Y391 of the G α 5 helix, forming a hydrophobic
interface with the G protein.

Taken together, our structural analysis supports the partial presence and conserved spatial
architecture of both the UQC and HLY motifs in ADGRL4, indicating functional relevance.
Given the structural conservation of these motifs across aGPCRs and their inferred roles in
receptor activation, experimental mutagenesis and signalling assays would indeed be
valuable for further validating their roles in ADGRL4. We are intending to do this work in the
context of a future publication that will include the inactive state structure of ADGRL4, where
we will focus on delineating the activation process. Given the scope and focus of the current
manuscript, and since our existing structural data provide initial mechanistic insight into these
motifs, we suggest that such functional mutagenesis is best addressed in our next publication.

We have added this analysis to Discussion (lines 298-307):

“Structural inspection of ADGRL4 revealed partial conservation of the upper
quaternary core (UQC) activation motif (Seufert et al., 2023) (F3.44, M3.47, W6.53),
where W6.53 is linked via Q7.49 to the conserved FLLG motif (Seufert et al., 2023)
stabilizing the TM6 kink. However, in contrast to other aGPCRs with the UQC motif
such as ADGRG2 and ADGRG3, W6.53 in ADGRL4 does not interact directly with the
tethered agonist. The HLY motif (Seufert et al., 2023) is also conserved, with L3.54
directly contacting the G α 5 helix. These structural features suggest partially analogous
activation mechanisms to those described for ADGRG2 and ADGRG3, although
experimental validation in future studies will be necessary to elucidate this.”

MINOR POINTS

Abstract + Introduction

“... are a 33-member subfamily of class B ...”. They are not a subfamily but a family. And in humans there are only 32, not 32 homologs (ADGRE4/EMR4 is a pseudogen in humans). As the authors refer to the human family, please correct also the number wherever it appears in the manuscript.

As requested, in both the abstract (line 16) and introduction (line 34) we have made the following changes: changed “subfamily” to “family”; and changed “33-member” to “32-member”. (Our original total (n=33) referred to both the protein-coding (n=32) and pseudogene (n=1) members).

38: when citing ref. 3 as initial discovery of the Stachel mechanism, also cite ref. 10.1073/pnas.1421785112

The new reference (Stoveken et al. DOI:10.1073/pnas.1421785112) has been added as reference 4 (line 43).

38: the receptor-integral Stachel is not a peptide, but part of the contiguous full-length or at least CTF receptor portion. Only when the Stachel is generated by peptide synthesis, e.g. to be added in pharmacological assays, the “Stachel” should be called a “Stachel peptide”

Thank you for the suggestion: we have changed “stachel peptide” to “stachel”.

The text (line 37-43) now reads:

“Most aGPCRs contain a GPCR autoproteolysis-inducing (GAIN) domain immediately before the first transmembrane helix (Fig. 1a), which possesses a hidden tethered agonist also known as the stachel³⁻⁶.”

42: There is more suitable reference to an original research paper instead of the review in ref. 6: 10.1016/j.molcel.2020.12.042

We have now replaced that reference at the end of line 43 (formerly: Langenhan et al. DOI:
[10.1126/scisignal.2003825](https://doi.org/10.1126/scisignal.2003825)) with the recommended new reference (Beliu et al. DOI:
[10.1016/j.molcel.2020.12.042](https://doi.org/10.1016/j.molcel.2020.12.042)).

45-47: The cited ref. 16 does not contain a 7TM but an ECR structure and is thus unsuitable
to validate the authors' statement: "This activation model is supported by recently solved
cryo-EM structures of active-state aGPCRs"

Thank you for this comment. We have now removed the former reference 16 (Kordon et al.
DOI: [10.1038/s41467-023-36312-7](https://doi.org/10.1038/s41467-023-36312-7)).

52-54: "In general, higher ADGRL4 expression within the tumour microenvironment across
the majority of these tumours correlates with more aggressive disease in vitro and in vivo".
Cite reference(s) supporting this claim.

The references have now been added to this sentence (line 57-59). It now reads:

"In general, higher ADGRL4 expression within the tumour microenvironment across
the majority of these tumours correlates with more aggressive disease in vitro and in
vivo(Abdul Aziz et al., 2016, Li et al., 2019, Guihurt Santiago et al., 2021, Sun and
Zhong, 2021, Sun et al., 2021, Wang et al., 2024b, Geng et al., 2024, Mao et al.,
2024, Wang et al., 2024a, Wang and Zhang, 2024)."

90 ff.

"Mutated Full-Length 1' (FLmut1) was the full-length construct containing the alanine
substitutions H405A and T407A at the GPCR proteolysis site (GPS), which prevented
92 autocatalytic cleavage and hence NTF and CTF dissociation".

- Ext. Data Fig. 3c shows a WB of all these mutants but not the L4 protein with intact GPS to
allow judgement if the WT receptor is autoproteolysed to start with. Include a full length WT
protein too and run it along with the mutants.

Thank you for this comment. The western blot appearance of wild-type full-length ADGRL4
has been previously published by the senior author's former laboratory. For examples, please
see Figure S5E in Masiero et al, 2013 (DOI: [10.1016/j.ccr.2013.06.004](https://doi.org/10.1016/j.ccr.2013.06.004)); Figure 1B in Favara
et al, 2019 (DOI: [10.3390/metabo9120287](https://doi.org/10.3390/metabo9120287)), and Figure 1A in Favara et al, 2021 (DOI:
[10.1038/s41598-021-85408-x](https://doi.org/10.1038/s41598-021-85408-x)).

In our current manuscript, Ext. Data Fig 3c reports the western blot of stable HEK293 GnTi-
TetR cell lines generated to express our structural studies constructs. We did not generate a
wild-type full-length ADGRL4 construct for structural studies because early preliminary tests
(unreported in this study as it used an earlier generation of purification tags) with full-length
ADGRL4 suggested that it was not stable enough to withstand dissociating into NTF and CTF
fragments when subjected to our membrane solubilisation and protein purification approach
reported in the manuscript. Thus, we generated a mutated full-length (FLmut) construct.
Unfortunately, the time required to generate a new wild-type full-length ADGRL4 construct with
similar tags to the current structural studies construct as well as generate the stable HEK293
GnTi- TetR cell line exceeds the time allocation allowed by the journal.

- It is puzzling that the human FL WT L4 protein should be ca. 78 kDa but the GPS-deficient
versions shown by the authors (FLmut2, FLmut2.1) run at double the size (see Ext. Fig. 3c).
Elaborate on this conflict between prediction and data.

- Similarly: L4-CTF (CTF2) should be 32 kDa but runs at appx. 60 (see Ext. Fig. 3c).

Elaborate on this conflict between prediction and data.

- Please indicate the expected sizes in kDa for all proteins (FL, CTF, CTF-SmBit fusions)
tested in WB so the reader can judge the depicted bands.

Thank you for these questions. The Uniprot reference amino acid sequence for wild-type full-
length ADGRL4 is indeed predicted to be 78 kDa, however this does not include 10x predicted
N-glycosylation sites (adding an approximate 25 kDa), nor the weight of our detection and
purification tags. For published evidence of wild-type full-length ADGRL4's glycosylation,
please see the western blot appearance of wild-type full-length ADGRL4 subjected with and
without PNGase treatment: Figure S5E in Masiero et al, 2013 (DOI:
10.1016/j.ccr.2013.06.004).

To address the molecular weight question, we have added the predicted molecular weights for
each construct to the figure legend of Extended Data Fig 3 (line 1554-1556):

"The predicted molecular weights for the assayed ADGRL4 constructs were: ~140kDa
for FLmut2 and FLmut2.1; ~74kDa for CTF2; ~100kDa for CTF2A; and ~103kDa for
CTF2B."

Line 176 ff.

Despite the L4-CTF 7TM structure at hand, the authors have forgone the chance to clearly

comment on the exceptional placement of L4 within the aGPCR family. Based on its
adhesive domains in the ECR it was classically considered to be an ADGRE member, but
the primary sequence of the 7TM domain placed it firmly within the ADGRL subfamily (e.g. in
10.1016/j.ygeno.2003.12.004 <https://pubmed.ncbi.nlm.nih.gov/15203201/> ,
10.1124/pr.114.009647 <https://pubmed.ncbi.nlm.nih.gov/25713288/>, 10.1111/nyas.14192
<https://pubmed.ncbi.nlm.nih.gov/31365134/>).

With the structure of the active 7TM in hand now, and the existence of atomistic structures of
other L and E 7TM domains to compare it to (L3 and E5), the authors should comment and
evaluate the kinship of L4 to the L and E subfamilies, generate superimposition images
similar to their Fig. 4. The integration of F1 in the comparisons there muddies the clarity of
the data presentation.

Could the authors make a clear statement, perhaps in the heading of the section explaining
the results in Fig. 4, to render their verdict on the similarity and placement of L4 between the
E and L subfamily clear?

The comparisons we performed in the manuscript had the explicit purpose of comparing only
G_q coupled receptors, which is why we included F1, and not to any phylogenetic relationships
between the receptors. We know from many other GPCR structures that there are differences
in a receptor when it is coupled to different G proteins, which is why it is essential to compare
receptors coupled to the same G proteins. For example, comparison of L4 with L3 gives
RMSDs of 1.0, 1.1, 1.2 or 1.4 Å depending upon whether L3 is coupled to either G_q , G_{12} , G_s
or G_i . The comparison requested by the referee between L4 and either L3 or E5, all receptors
coupled to G_q , give identical RMSDs of 1.0 Å. This comparison therefore does not add any
further information to the categorisation of L4 and no statement in the text is necessary.

Fig. 1

Fig. 1d-h: although the caption states that three biological replicates were obtained each
containing 3 technical replicates (i.e. 9 data points), these 9 data points are only plotted in
Fig. 1d. In Fig. e-h only three datapoints are visible. What do they correspond to and why
does the data presentation differ from the one in Fig. 1d?

Thank you for this question. Initially, Fig 1e-h only showed $n=1$ biological replicates as stated
in the figure legend, because this was reporting negative results.

However, we have now repeated experiments so that all negative signalling assays have
been performed $n=3$ times (3 biological replicates). Now, all graphs within Fig 1e-h have 9

data points. We have also updated the statistics reported in Extended Data Table 1 in light of
these new replicates.

Generally, the authors should add their statistical result comparing FL1 to CTF1 and FLmut1
to CTF1 coupling to the intact G protein (just as annotated in Fig. 1d) in Fig. 1e-g even
though the results indicated no significant difference.

This has now been completed as suggested and all graphs within Fig 1e-g have undergone
the same comparisons as Fig 1d. We have also updated the statistics reported in Extended
Data Table 1 in light of these additional comparisons.

Ext. Data Fig. 1

Ext. Data Fig. 1a: although the CTF1 variant was N-terminally tagged with an HA tag it did
not show at the cell surface in neither combination with or without and intracellular pathway
pathway component. This is observation has been made for other aGPCRs before, while
others reported that CTF receptor variants can traffick to the plasma membrane and can get
detected there. Please describe this result and comment on your conclusions in the text
where you refer to the data, i.e. in the block 108-111. Could the coupling of L4-CTF1 to Gq
be greater than of L4-FL because the subcellular location of the receptor is different? Could
the receptor be internalised or rapidly get internalised because the NTF is missing? This is a
particular possibility that the authors considered for their L4-CTF2 variant later introduced
and described in the manuscript, where they write in main text: "ADGRL4-CTF2 had the
lowest with close to no surface expression suggesting either instability, misfolding and/or
rapid turnover of the potentially constitutively active receptor" (lines 142-144).

So, if this applies to the L4-CTF1 variant too, do the authors assume that their Gq coupling
to L4 -CTF protein occurs in the ER or en route to the plasma membrane, as opposed to at
the cell surface? Is this resonable?

Thank you for this interesting question regarding Ext. Data Fig. 1a. We would like to highlight
that the HA CTF1 variant was expressed at the cell surface as detected by flow cytometry,
with an average cell-surface expression of 5% (Extended Data Fig. 1a, and Source Data.xls
file). The question regarding subcellular localisation of the CTF variants is an important and
interesting one, but is beyond the scope of this structural biology paper. We hope to
investigate this in future work.

The following text has been added as requested (line: 124-128):

“Representative mean cell surface expression of CTF1 (5%) was significantly lower than for FL1 and FLmut1 (88%) (Extended Data Fig. 1a, and Source Data file) suggesting either rapid cell-surface internalisation of activated receptor, expression and activity at subcellular locations, or protein misfolding. This difference was not seen for whole-cell expression where representative results showed marginally higher expression for CTF1 than FL1 and FLmut1.”

Reviewer #2 (Remarks to the Author):

The current study provides the structure of the C-terminal fragment of the adhesion GPCR, ELTD-1. ELTD-1 has roles in cancer and angiogenesis, so understanding the structural basis of its activation mechanism and G protein coupling is significant. Overall the structural work is of high quality and demonstrates that ELTD-1 is activated by its tethered agonist. The pose of the tethered agonist within the 7TM orthosteric resembles the poses of tethered agonists for other adhesion GPCRs with recently solved structures. The structural and signalling data provide a potential explanation of why ELTD-1 has been considered a poor G protein coupler or lacked this ability altogether. The ELTD-1 structure was of a complex with mini-Gq heterotrimer. The position of the Gaq alpha-5 helix that engaged the 7TM was in a pose that suggested weaker Gq coupling and did not resemble other active-state GPCR structures in which this helix is inserted more deeply into the intracellular opening of the 7TM.

We thank the reviewer for their clear assessment of our work. We have responded to issues raised below in blue text. Line numbers below refer to the version of the manuscript with Track Changes visible.

There are some potential caveats of the signaling data that require additional experimentation or explanation. Most concerning is the use of N-terminal HA tag ELTD-1 expression constructs. For evaluation of signaling from the full length wildtype and cleavage defective mutants, this tag is probably not an issue, but for evaluation of the CTF constructs it certainly is. The original study and many subsequent studies that describe the adhesion GPCR tethered agonist mechanism demonstrate clearly that when any sequence other than an initiator methionine is appended onto the N-terminus of adhesion GPCR CTFs, that this abrogates tethered agonism. The series of papers in 2022 from multiple labs that described several adhesion GPCR tethered agonist-activated structures capitalized on this finding and

393 utilized the “naked” methionine CTF constructs. It was presumed with good assumption that
the methionine was cleaved off by methionyl aminopeptidase as is the case for most cellular
proteins and that no density for a N-terminal methionine was observed in the structures. So,
if an HA-tagged CTF construct was used throughout the signaling studies of Figure 1, this
requires additional examination with a non-tagged construct.

We thank the reviewer for their comment. We disagree that the addition of an HA-tag to the
N-terminal end of the CTF construct is problematic as it has been successfully performed for
a number of adhesion GPCRs when reporting the signalling ability of their CTF variant. This
list includes: ADGRD1 and ADGRG6 (Liebscher et al, 2014; DOI:
10.1016/j.celrep.2014.11.036)), ADGRG2 (Demberg et al, 2015; DOI:
10.1016/j.bbrc.2015.07.020) and ADGRG5 (Wilde et al, 2016; DOI: 10.1096/fj.15-276220).
Additionally, other N-terminal tags have been successfully used in CTF variants when
reporting signalling ability, such as N-terminal FLAG-tag for the CTF variant of GPR116 (Brown
et al, 2017; DOI: 10.1172/jci.insight.93700). We thus stand by our results and experimental
approach.

The aim of our paper was to solve the active-state high-resolution structure and solving the
signalling was a key step in this process. Whilst our paper was under review, a new paper was
published reporting the first two ADGRL4 ligands as well as downstream evidence of G_q
coupling using the NFAT assay (Amisten et al, 2025 (DOI: 10.1016/j.bbrc.2025.151785). This
adds further weight to our signalling results.

The following has now been added to the results section (line 107-110):

“The addition of the N-terminal HA tag to a C-terminal Fragment (CTF) aGPCR has
been previously used as a detection tag in aGPCR signalling experiments and has
not interfered with signalling by a number of CTF aGPCRs, including ADGRD1 and
ADGRG6³, ADGRG2⁴⁸ and ADGRG5⁴⁹.”

Additionally, the following has now been added to the discussion section (lines 275-278):

“ Whilst this paper was under review, another group has corroborated our G_q findings
by means of an NFAT-reporter assay (downstream of G_q) in U87 glioblastoma cells
(Amisten et al., 2025). This suggests that cellular context plays an important role as
previous ADGRL4 NFAT-reporter studies in HEK293 cells were negative (Favara et
al., 2021, Dates et al., 2024)”

Related to the signalling, we performed alanine mutagenesis experiments during this revision
in response to reviewer 1. Scanning alanine mutagenesis was performed on amino acid
residues within our cryo-EM model's tethered agonist region which formed contacts with the
orthosteric binding pocket within ≤ 3.9 Å. Results show that seven residues within ADGRL4's
tethered agonist region contribute to activation (H408, F409, I411, L412, M413, S414, S415),
the most critical being the hydrophobic M413 and F409 residues. The following new figures
and table have been added to the manuscript to report the above tethered agonist
mutagenesis results: Fig. 3e; Extended Data Fig. 5; Extended Data Table 3. New text
regarding the mutagenesis experiments has been added to the results section (line 216-230),
discussion section (line 291-296) and methods section (line 568-576).

One additional point that the authors do touch on is the odd results about positive G_o
coupling that is statistically significant, but is brushed off because the coupling appeared
stronger to the full length receptor and this does not fit the model. The statistically significant
G_q coupling to the CTF construct did fit the model, so this was accentuated in the write-up.
Is there any assay problem with mini- G_o ? Mini- G_o in a 2021 GPR97 structural study was
modified with a non-native palmitate on its alpha 5 helix, providing a completely artificial
mode of receptor / G protein engagement.

We can confirm that the mini- G_o used in our experiment did not contain a non-native palmitate
on its alpha 5 helix and that the G_o assay worked reporting an approximate 5 fold change
difference between positive and negative controls. In this revision we have repeated to $n=3$
biological replicates all the signalling results previously reported as $n=1$ in Fig1e-h. For G_o , the
updated results remain the same. This fold change difference between positive and negative
G_o controls was roughly similar across other signalling assays reported in Fig 1d-g.

Nonetheless, the results of the positive G_o coupling are indeed interesting and warrant further
investigation in native cell systems, but are beyond the scope of this structural biology paper.
Our aim was to solve the high-resolution structure of active-state ADGRL4 which required
finding the best G protein coupler, and this is exactly what we did.

For the structural studies, it is not completely clear whether the issue with modified N-termini
pertains. Four different expression constructs for use in receptor preparation for structural
determination are described, and it might be the case that cleavage of the engineered
protease sites produced an authentic N-terminus. This should be better clarified in the
manuscript. How did the structure(s), which begin with the threonine of the authentic

tethered agonist arise?

We solved our high-resolution CTF structure without resorting to cleaving off any N-terminal
purification tags which were appended to the N-terminal aspect of the tethered agonist by a
flexible GS linker as described in the methods and results. Although we had engineered
protease sites in the purification tags, we did not cleave them in the structure solved by cryo-
EM and presented in our manuscript. Due to them being disordered and mobile, cryo-EM did
not detect any tags located N-terminal to the linkered tethered agonist region.

We have added the following line to the methods section to clarify this (line 504-505):

“The engineered protease sites were not cleaved off and the entire purified
recombinant protein (including all purification and detection tags) was processed.”

**Reviewer #3 (Remarks to the Author):**

This study by Favara and colleagues presents presents a comprehensive investigation into
the signaling and structural properties of ADGRL4, an orphan adhesion GPCR that has been
implicated in a variety of cancers. Despite its clinical significance, ADGRL4 has remained
poorly understood, and previous studies failed to detect its coupling to heterotrimeric G
proteins, leading to uncertainty about its signaling mechanisms. In this study, the authors
use a highly sensitive NanoBiT-based bioluminescence assay to demonstrate that ADGRL4
exhibits weak but specific coupling to the Gq protein, while no robust interaction is observed
with other G protein subtypes (Gs, G12, Go) or with β -arrestin 1/2. This weak coupling,
which was previously undetectable using standard approaches, is further supported by cryo-
EM structural analysis. The authors then determined the structure of ADGRL4 in complex
with mini-Gq at an overall resolution of 3.1 Å. The structure reveals that ADGRL4 maintains
the canonical architecture of other aGPCRs and is activated by its endogenous tethered
agonist, which adopts a short α -helical conformation and binds within the orthosteric pocket.
Notably, the interface between ADGRL4 and Gq exhibits markedly fewer and weaker
contacts compared to other structurally characterized Gq-coupled aGPCRs, providing a
structural rationale for the observed weak signaling output.

We thank the reviewer for their precise assessment of our work. We have replied to each
point raised below in blue text. Line numbers below refer to the version of the manuscript
with Track Changes visible.

1. A major concern is whether the Gq-coupling activity of ADGRL4 demonstrated in this
study accurately reflects its physiological signaling. The mini-G protein constructs used are
highly engineered, and the signaling assays rely on overexpression of both the receptor and
G proteins in heterologous systems. It remains unclear whether ADGRL4 can activate
endogenous Gq signaling in native cells. As the authors mentioned, the JAK/STAT3 and
MAPK/ERK pathways can be driven by various upstream signaling molecules. To address
this, it is important to assess whether the CTF1 construct can stimulate wild-type Gq
signaling by measuring calcium mobilization or IP3 accumulation. Ideally, these experiments
should be performed in native cells without overexpression of G proteins. However, even
studies in heterologous systems with co-expression of CTF1 and wild-type Gq would provide
critical supporting evidence.

We agree that the mini-G protein constructs used are highly engineered. Whilst our
manuscript was in review, a paper was published supporting our findings by reporting
evidence of ADGRL4 G_q coupling as evidenced by an NFAT-reporter assay in U87
glioblastoma cells (Amisten et al, 2025 (DOI: 10.1016/j.bbrc.2025.151785). This has been
added and referenced in the discussion (lines 275-278) as follows:

“ Whilst this paper was under review, another group has corroborated our G_q findings
by means of an NFAT-reporter assay (downstream of G_q) in U87 glioblastoma cells
(Amisten et al., 2025). This suggests that cellular context plays an important role as
previous ADGRL4 NFAT-reporter studies in HEK293 cells were negative (Favara et
al., 2021, Dates et al., 2024)”

The question regarding ADGRL4's signalling in native cells and its exact relationship with the
JAK/STAT3 and MAPK/ERK pathways in these cells is important and is interesting, but is
beyond the scope of this structural biology paper. Our aim was to solve the high-resolution
structure of ADGRL4 in the active-state which required us to determine the best G protein for
it to couple with in order to stabilise the receptor in a fully active state. We plan to study
native cell systems to elucidate signalling networks for a future paper.

2. CTF1 is supposed to be a constitutively active GPCR. However, in Fig. 1h, why CTF1 is
less active in b-arrestin recruitment compared to FL1 and FLmut1?

Thank you for this question. Figure 1h shows that the CTF1 version of ADGRL4 does not
activate β-arrestin 1 or 2 as it has a lower signal than the FL1 and FLmut1 (FLmut1 is
hypothesised to be inactive due to its GPS-site mutation). The reason for the lower CTF1
level may be related to differences in cell surface expression as seen in Extended data

figure 1a. Finally, not all activated GPCRs activate arrestins and this is likely to be the case
for ADGRL4.

3. There is ongoing debate in the adhesion GPCR field regarding whether all aGPCRs, or
how many, are activated by Stachel peptides and whether cleavage of the NTF necessarily
leads to its dissociation from the CTF. In some cases, Stachel peptides may not serve as
activators, or they may represent only one of multiple activation mechanisms under
physiological conditions. Although the cryo-EM structure clearly shows that the Stachel
peptide binds to ADGRL5 in a mode similar to that observed in other aGPCR structures, this
does not necessarily establish it as the endogenous agonist. For instance, there are
structures of opioid receptor-G protein signaling complexes bound to an inverse agonist
(<https://doi.org/10.1038/s41589-024-01812-0>), suggesting that structural engagement alone
does not confirm physiological agonism. It is therefore recommended to test the free form of
the Stachel peptide in the NanoBIT-based G protein recruitment assays. Have any studies
examined the function of the free Stachel peptide, e.g. whether it can stimulate the
JAK/STAT3 or MAPK/ERK pathways via ADGRL4?

This is a very interesting question. The senior author has previously synthesised ADGRL4's
tethered agonist region as peptides of increasing length (for published evidence, see Figure
2A from Favara et al, 2021; DOI: 10.1038/s41598-021-85408-x). Unfortunately, these
ADGRL4 tethered agonist peptides were extremely hydrophobic and would not dissolve
unless they were in very high non-physiological DMSO concentrations, a problem which was
also noted as well by the two collaborating laboratories involved in that paper (Heptares
Therapeutics UK; and the laboratory of Prof Ines Liebscher, Leipzig). Modifications with water-
soluble molecules (biotin, PEG2, betaine) as well as with water soluble amino acids did not
improve solubility (see Favara et al, 2021; DOI: 10.1038/s41598-021-85408-x). This is in
contrast to the peptides of the tethered agonist of GPR116 (Brown et al, 2017; DOI:
10.1172/jci.insight.93700) which in the senior author's experience were a lot more water
soluble. Therefore, whilst the suggested experiment is interesting, it is not possible to pursue
due to insolubility issues.

4. As the authors noted, several structures of adhesion GPCRs bound to their tethered
peptides have been reported, all of which exhibit a similar ligand-binding mode. In fact,
AlphaFold3 can predict an active-state structure of ADGRL4 bound to its tethered agonist
peptide, which closely resembles the experimentally determined structure reported in this
study (RMSD < 1 Å for C α atoms and ~2 Å for all atoms). While this does not diminish the
value of the experimentally resolved structure, especially considering that it provides more

information on receptor signaling, it would be informative to include a direct comparison with
the AlphaFold3 model and discuss the implications of AI-based structure prediction for future
drug development targeting ADGRL5. It may also be worth discussing whether AI tools can
accurately predict an inactive conformation of ADGRL5.

Thank you for your suggestion. Within lines 330-331 in the discussion we reference a range
of key papers and approaches (including AI *in-silico* approaches) for screening and
developing tool compounds for potential therapeutic use using our experimentally-solved
structure. As regards the AlphaFold3 comparison, we agree that the similarity between the
predicted model and the experimentally determined structure is quite good, but given the
choice between starting a £10k programme of *in silico* screening using either a cryo-EM
structure or an AI-derived model, we would definitely use the cryo-EM structure. We have
also been working on making a model to the inactive state and here the AlphaFold model is
very poor (in our view) and we are having to do significant rebuilding based on the inactive
L3 structure. Given these caveats, we feel it is unnecessary to discuss AlphaFold models in
our manuscript.

5. To enable reproducibility, it is recommended to include more detailed information on the
design of the signaling constructs, specifically, the exact residue ranges of ADGRL5 used in
each construct. For people outside the adhesion GPCR field, the start of the Stachel
sequence may not be readily apparent, so clarifying this would improve accessibility and
help other researchers to reproduce the data.

We have now included residue range of ADGRL4 used in each construct in the method
section based on the UNIPROT reference.

The following text has been added to the method section (374-377)

"Using the Uniprot ADGRL4 reference sequence (Q9HBW9), untagged Full-Length
ADGRL4 constructs comprised amino acids K2-R690, whilst untagged C-terminal
Fragment constructs comprised amino acids T407-R690."

6. It is recommended to revise the title. Many GPCRs are implicated in cancer. This paper
does not explore the functional roles of ADGRL4 in cancer. And it is not a well established
cancer target.

We disagree that ADGRL4 is not a well-established cancer target. There is a growing
experimental literature spanning both animal and human models across many tumour-types

reporting its relevance as a potential tumour-associated prognostic marker, its functional
importance in tumourigenesis, and its potential for therapeutic targeting. Examples of tumours
where ADGRL4 is relevant have been cited in our paper and include glioblastoma (Dieterich
et al., 2012, Towner et al., 2013, Li et al., 2019, Wang et al., 2024b), gastric cancer (Sun and
Zhong, 2021, Wang et al., 2024a), colorectal cancer (Masiero et al., 2013, Abdul Aziz et al.,
2016, Sun et al., 2021), breast cancer (Sheldon et al., 2021), renal cancer (Masiero et al.,
2013, Wang and Zhang, 2024), ovarian cancer (Masiero et al., 2013), hepatocellular
carcinoma (Kan et al., 2018, Mao et al., 2024), cholangiocarcinoma (Li et al., 2024),
retinoblastoma (Guihurt Santiago et al., 2021) and neuroblastoma (Geng et al., 2024).

Additionally in both their summaries, both reviewer 1 and reviewer 2 remark on ADGRL4's role
in cancer biology. Reviewer 1 writes that "The role of L4 in cancer biology underlines the
importance of this manuscript, as indeed the structure may lend themselves to expedited
development and testing of compounds hedging in the activity of L4 in cancers". Reviewer 2
writes that "ELTD-1 has roles in cancer and angiogenesis, so understanding the structural
basis of its activation mechanism and G protein coupling is significant".

Finally, our title does not imply an exploration of functional roles in cancer. We hope that these
clarifications justify our argument to retain the original title.

**References:**

- Abdul Aziz, N. A., Mokhtar, N. M., Harun, R., Mollah, M. M., Mohamed Rose, I., Sagap, I.,
Mohd Tamil, A., Wan Ngah, W. Z. & Jamal, R. 2016. A 19-Gene expression signature
as a predictor of survival in colorectal cancer. *BMC Med Genomics*, 9, 58.
- Amisten, S., Rezeli, M., Grossi, M., Bryl-Gorecka, P., Håkansson, A., Torngren, K., Marko-
Varga, G., Erlinge, D. & Olde, B. 2025. The DNA repair factor ku80 binds and activates
the adhesion receptor ELTD1/ADGRL4. *Biochem Biophys Res Commun*, 764, 151785.
- Dates, A. N., Jones, D. T. D., Smith, J. S., Skiba, M. A., Rich, M. F., Burruss, M. M., Kruse, A.
C. & Blacklow, S. C. 2024. Heterogeneity of tethered agonist signaling in adhesion G
protein-coupled receptors. *Cell Chem Biol*, 31, 1542-1553.e4.
- Demberg, L. M., Winkler, J., Wilde, C., Simon, K. U., Schon, J., Rothmund, S., Schoneberg,
645 T., Promel, S. & Liebscher, I. 2017. Activation of Adhesion G Protein-coupled
Receptors: AGONIST SPECIFICITY OF STACHEL SEQUENCE-DERIVED
PEPTIDES. *J Biol Chem*, 292, 4383-4394.
- Dieterich, L. C., Mellberg, S., Langenkamp, E., Zhang, L., Zieba, A., Salomäki, H., Teichert,
649 M., Huang, H., Edqvist, P. H., Kraus, T., Augustin, H. G., Olofsson, T., Larsson, E.,
Söderberg, O., Molema, G., Pontén, F., Georgii-Hemming, P., Alafuzoff, I. & Dimberg,
651 A. 2012. Transcriptional profiling of human glioblastoma vessels indicates a key role
of VEGF-A and TGFβ2 in vascular abnormalization. *J Pathol*, 228, 378-90.
- Favara, D. M., Liebscher, I., Jazayeri, A., Nambiar, M., Sheldon, H., Banham, A. H. & Harris,
654 A. L. 2021. Elevated expression of the adhesion GPCR ADGRL4/ELTD1 promotes
endothelial sprouting angiogenesis without activating canonical GPCR signalling. *Sci*
*Rep*, 11, 8870.

Geng, G., Zhang, L., Yu, Y., Guo, X., Li, Q. & Ming, M. 2024. ADGRL4 Promotes Cell Growth,
Aggressiveness, EMT, and Angiogenesis in Neuroblastoma via Activation of
ERK/STAT3 Pathway. *Curr Mol Med*.

Guihurt Santiago, J., Burgos-Tirado, N., Lafontaine, D. D., Mendoza Sierra, J. C., Camacho,
R. H., Vecchini Rodríguez, C. M., Morales-Tirado, V. & Flores-Otero, J. 2021. Adhesion
G protein-coupled receptor, ELTD1, is a potential therapeutic target for retinoblastoma
migration and invasion. *BMC Cancer*, 21, 53.

Kan, A., Le, Y., Zhang, Y. F., Duan, F. T., Zhong, X. P., Lu, L. H., Ling, Y. H. & Guo, R. P. 2018.
ELTD1 Function in Hepatocellular Carcinoma is Carcinoma-Associated Fibroblast-
Dependent. *J Cancer*, 9, 2415-2427.

Li, J., Shen, J., Wang, Z., Xu, H., Wang, Q., Chai, S., Fu, P., Huang, T., Anas, O., Zhao, H., Li,
668 J. & Xiong, N. 2019. ELTD1 facilitates glioma proliferation, migration and invasion by
669 activating JAK/STAT3/HIF-1 α signaling axis. *Sci Rep*, 9, 13904.

Li, Z., Nguyen Canh, H., Takahashi, K., Le Thanh, D., Nguyen Thi, Q., Yang, R., Yoshimura,
671 K., Sato, Y., Nguyen Thi, K., Nakata, H., Ikeda, H., Kozaka, K., Kobayashi, S., Yagi, S.
& Harada, K. 2024. Histopathological growth pattern and vessel co-option in
intrahepatic cholangiocarcinoma. *Med Mol Morphol*, 57, 200-217.

Mao, D., Wang, H., Guo, H., Che, X., Chen, M., Li, X., Liu, Y., Huo, J. & Chen, Y. 2024.
Tanshinone IIA normalized hepatocellular carcinoma vessels and enhanced PD-1
inhibitor efficacy by inhibiting ELTD1. *Phytomedicine*, 123, 155191.

Masiero, M., Simoes, F. C., Han, H. D., Snell, C., Peterkin, T., Bridges, E., Mangala, L. S., Wu,
S. Y., Pradeep, S., Li, D., Han, C., Dalton, H., Lopez-Berestein, G., Tuynman, J. B.,
Mortensen, N., Li, J. L., Patient, R., Sood, A. K., Banham, A. H., Harris, A. L. & Buffa,
F. M. 2013. A core human primary tumor angiogenesis signature identifies the
endothelial orphan receptor ELTD1 as a key regulator of angiogenesis. *Cancer Cell*,
24, 229-41.

Ping, Y.-Q., Xiao, P., Yang, F., Zhao, R.-J., Guo, S.-C., Yan, X., Wu, X., Zhang, C., Lu, Y., Zhao,
F., Zhou, F., Xi, Y.-T., Yin, W., Liu, F.-Z., He, D.-F., Zhang, D.-L., Zhu, Z.-L., Jiang, Y.,
Du, L., Feng, S.-Q., Schöneberg, T., Liebscher, I., Xu, H. E. & Sun, J.-P. 2022.
Structural basis for the tethered peptide activation of adhesion GPCRs. *Nature*.

Seufert, F., Chung, Y. K., Hildebrand, P. W. & Langenhan, T. 2023. 7TM domain structures of
adhesion GPCRs: what's new and what's missing? *Trends Biochem Sci*, 48, 726-739.

Sheldon, H., Bridges, E., Silva, I., Masiero, M., Favara, D. M., Wang, D., Leek, R., Snell, C.,
Roxanis, I., Kreuzer, M., Gileadi, U., Buffa, F. M., Banham, A. & Harris, A. L. 2021.
ADGRL4/ELTD1 Expression in Breast Cancer Cells Induces Vascular Normalization
and Immune Suppression. *Mol Cancer Res*, 19, 1957-1969.

Sun, B. & Zhong, F. J. 2021. ELTD1 Promotes Gastric Cancer Cell Proliferation, Invasion and
Epithelial-Mesenchymal Transition Through MAPK/ERK Signaling by Regulating CSK.
*Int J Gen Med*, 14, 4897-4911.

Sun, J., Zhang, Z., Chen, J., Xue, M. & Pan, X. 2021. ELTD1 promotes invasion and
metastasis by activating MMP2 in colorectal cancer. *Int J Biol Sci*, 17, 3048-3058.

Towner, R. A., Jensen, R. L., Colman, H., Vaillant, B., Smith, N., Casteel, R., Saunders, D.,
Gillespie, D. L., Silasi-Mansat, R., Lupu, F., Giles, C. B. & Wren, J. D. 2013. ELTD1, a
potential new biomarker for gliomas. *Neurosurgery*, 72, 77-90; discussion 91.

Wang, L., Xu, Y., Jiang, L. & Liu, J. 2024a. The elevated expression of ADGRL4 indicates poor
prognosis in gastric cancer. *Asian J Surg*.

Wang, Q., Zhang, C., Pang, Y., Cheng, M., Wang, R., Chen, X., Ji, T., Yang, Y., Zhang, J. &
Zhong, C. 2024b. Comprehensive analysis of bulk, single-cell RNA sequencing, and
spatial transcriptomics revealed IER3 for predicting malignant progression and
immunotherapy efficacy in glioma. *Cancer Cell International*, 24, 332.

Wang, Z. & Zhang, Z. 2024. Single-cell analysis reveals ADGRL4+ renal tubule cells as a
highly aggressive cell type in clear cell renal cell carcinoma. *Sci Rep*, 14, 2407.

Xiao, P., Guo, S., Wen, X., He, Q.-T., Lin, H., Huang, S.-M., Gou, L., Zhang, C., Yang, Z.,
Zhong, Y.-N., Yang, C.-C., Li, Y., Gong, Z., Tao, X.-N., Yang, Z.-S., Lu, Y., Li, S.-L., He,

711 J.-Y., Wang, C., Zhang, L., Kong, L., Sun, J.-P. & Yu, X. 2022. Tethered peptide
activation mechanism of the adhesion GPCRs ADGRG2 and ADGRG4. *Nature*.

We thank the reviewers for their second detailed assessment of our work and their further
recommendations for improving the manuscript. We have now ensured that all reviewer
requests have been implemented. Their suggestions have greatly improved the manuscript
and for this we are grateful.

Below are our replies to all comments in blue (all line numbers referenced in our replies to the
reviewers refer to the PDF manuscript with Track Changes visible [manuscript_tracked.pdf]).
All additions to our manuscript are visible as Tracked Changes.

Below is a summary of all major figure and table revisions to our manuscript:

- 11 A. We have completed all mutagenesis as requested by reviewer 1 and 2. We have added
additional paragraphs on limitations as requested by reviewer 3.
- B. New Figure 5 containing 3 panels (reports 7TM activation motif mutagenesis
experiments). The previous Figure 5 has now been renamed to Figure 6.
- C. Additional panels added to Extended Data Figure 1 (two extra panels: d,e) and
Extended Data Figure 5 (two extra panels: b,c)
- D. New Extended Data Table 4 added (corresponding to the data in the new Figure 5 and
new panels in Extended Data Figure 5)
- E. Additional data added to Extended Data Table 1 from the requested 'HA construct
versus Δ HA construct' G protein coupling experiments.
- F. New Supplementary Figure 2 added (corresponding to the aGPCR family-wide
sequence alignment performed as a precursor to the 7TM activation motif
experiments).
- G. Methods, discussion and results have been updated to account for the above
additional experiments.
- H. Changes to previous figures:
- • Figure 4 legend: changes made to text to enhance clarity.
 - • Figure 6c: colouring improved for clarity. Figure 6 legend: changes made to
text to enhance clarity.

**Reviewer #1 (Remarks to the Author):**

This is a revised version of a manuscript reporting the G protein bound structure of L4-CTF
and its function as a Gq coupled receptor. Most of my comments were addressed one
regarding the activation motifs was not.

**MAJOR POINTS**

ORIGINAL REQUEST

Therefore, the authors should capitalise on their newly identified contacts of the Stachel and
the binding pocket as reported in Fig. 3b,c, mutagenise the contacting residues on the
respective Stachel or 7TM side of the interface, and test mutant L4-NTF1 proteins for their
basal activity with their L4-CTF1-SmBit1/LgBit-Gq assay akin to similar binding relationship
studies reported for Gpr133 (e.g. see Fig. 2h [Stachel residues mutagenesis] and I [pocket
residue mutagenesis] in 10.1038/s41586-022-04619-y), Gpr56 (e.g. see Fig. 3c in
10.1038/s41586-022-04575-7), or Lphn3 (e.g. see Fig. 3d in 10.1038/s41586-022-04575-7).

COMMENT

I commend the authors on the results obtained with agonist mutagenesis confirming the
relevance of the active L4 structure.

We thank the reviewer for the positive feedback.

ORIGINAL REQUEST

The now available L4 signalling assay lends itself to clarify the presence or absence of
previously identified activation motifs within the 7TM domain of other aGPCRs (i.e. the upper
quaternary core, P/F/W/LφφG, H(N)L(M)Y, reviewed in 10.1016/j.tibs.2023.05.007). The
authors are asked first determine whether any of the previously characterised signaling
motifs are present in the 7TM domain of L4. If so, they should mutagenise them and test
them for changes in Gq coupling.

COMMENT

The authors looked for the presence of the activation motifs in L4, discovered some, but did
not test their functional relevance with their signalling assay. Instead they referred that to a
future manuscript (as for several other requests made by all three referees).

While I understand the time and resources required for these assays to run I would like to
emphasise the need to connect structural insights to functional data, in particular as the few
manuscripts that have done that in the past are still rare.

In order for the manuscript, to be published in Nature Communications, to add to the general
knowledge on how adhesion GPCRs principally work, the current principles such as the
found activation motifs need to be tested experimentally. Currently, the manuscript fails to do
this and only provides high-quality details about an individual receptor of this family.

I would thus ask the authors again to supply signalling tests of L4 variants containing
mutagenised UQC, FLLG and HLY motifs, when they have ascertained that the mutated
CTF proteins are expressed similarly to the WT L4-CTF.

Also: when the authors refer to the activation motifs in the text they should not cite Seufert et
al., 2023, which is a review of the state-of-the-art at the time containing all the references to
primary literature. I pointed this paper out to the authors to make it easier for them to get an
overview of the current state of knowledge on such motifs. But the authors should of course
rather cite the original work that discovered the motifs.

We thank the reviewer for this suggestion which we have now undertaken in full. This
additional set of experiments investigating putative 7TM activation motifs has strengthened
our manuscript and has generated interesting results. A full summary of results is presented
below.

The following new figures and table have been added to the manuscript to report these
results:

- 1. New Figure 5 (containing 3 panels) (lines 1111-1128). The previous Figure 5 has been
renamed Figure 6 (lines 1130-1142)
- 2. Additional panels (b and c) added to Extended Data Figure 5 (lines 1247-1262)
- 3. New Extended Data Table 4 added (corresponding to the new mutagenesis data in
Figure 5b and Extended Data Figure 5b-c) (line 1155 for the table)
- 4. New Supplementary Figure 2 (corresponding to the aGPCR family-wide sequence
alignment performed as a precursor to the 7TM activation motif experiments) (lines
1281-1287)

The following text has been added to the manuscript to describe these experiments and
results.

Within the **results section** the following text has been added (line 264-303):

Published aGPCR structures have proposed three putative core activation motifs
within the seven transmembrane (7TM) core: 1) the upper quaternary core motif (UQC;
F^{3.44}; M^{3.47}; F^{5.43}/F^{5.47}; W^{6.53})(Wootten residue numbering¹) as reported for ADGRG2

and ADGRG3²⁻⁵ (W^{6.53} has a hydrogen bond with Q^{7.49}); 2) the hydrophobic
P/F/W/L $\phi\phi$ G motif (where ϕ represents a hydrophobic residue) as reported for
ADGRF1 and ADGRD1^{3,4,6}; and 3) the H(N)L(M)Y motif as reported in ADGRG3 and
ADGRD1²⁻⁴. These are summarised in the review by Seufert et al⁷. We assessed
whether ADGRL4 contained any of these putative 7TM activation motifs by performing
sequence alignment against all human protein-coding aGPCRs and by comparing the
relative residue positions in the abovementioned aGPCRs with associated canonical
7TM activation motifs, with the structure of ADGRL4. Results (Fig.5a, Supplementary
Fig. 2) show that all three putative 7TM core activation motifs are present in ADGRL4,
albeit with some differences. The UQC motif in ADGRL4 comprises F505^{3.44}, M508^{3.47}
and W631^{6.53}, with a hydrogen bond between W631^{6.53} and Q656^{7.49}. The
F^{5.43}/F^{5.47} component of the canonical UQC motif is absent in ADGRL4. The
P/F/W/L $\phi\phi$ G motif in ADGRL4 comprises F625^{6.47}, L626^{6.48}, L627^{6.49}, and G628^{6.50}
(F $\phi\phi$ G); and its H(N)L(M)Y motif comprises H514^{3.53}, L515^{3.54} and Y516^{3.55}. When
comparing putative activation motifs across all aGPCRs, the UQC motif appears to
have the highest level of sequence conservation suggesting functional relevance
(Fig.5a).

Alanine scanning mutagenesis was performed on all three putative 7TM activation
motifs to interrogate functional effect (a total of 11 mutants were generated). Mutations
were introduced into the constitutively active ADGRL4-CTF1 construct with G_q
coupling assessed using the NanoLuc complementation assay. All mutants were
expressed similarly to wild type ADGRL4-CTF1 as determined by cell surface anti-HA
flow cytometry (Extended Data Fig. 5b). All the mutants exhibited a significant
decrease in G_q coupling when compared to wild type L4-CTF (Fig.5b, Extended Data
Fig.5c, Extended Data Table 4). Mutation of any of the amino acid residues in the UGC
abolished G_q recruitment. Mutations of two amino acid positions in each of the F $\phi\phi$ G
and H(N)L(M)Y motifs (L627A^{6.49} and G628A^{6.50}, and L515A^{3.54} and Y516A^{3.55},
respectively) also abolished G_q recruitment, although slightly lesser effects were
observed at residues F625A^{6.47} and L626A^{6.48} in the F $\phi\phi$ G motif and for H514A^{3.53} in
the H(N)L(M)Y motif. (Fig.5b, Extended Data Fig.5c, Extended Data Table 4). In
summary, these results provide the first evidence that ADGRL4 contains all three 7TM
activation motifs (Fig.5c) and that these all play an important role in ADGRL4 activation
and function, with loss of the UQC having the greatest negative effect.

Within the **discussion section**, the following has been added which replaces previous text
discussing putative 7TM activation motifs (line 383-400):

Our investigation of putative 7TM activation motifs within ADGRL4 shows that all three
previously reported hydrophobic activation motifs (UQC, F $\phi\phi$ G and H(N)L(M)Y motifs)
are present in ADGRL4 and are functionally important for ADGRL4 activation, with
mutagenesis of the UQC motif residues having the largest negative effect on receptor
activity. The UQC motif, canonically defined as F3.44, M3.47, F5.43/F5.47, and
W6.53, is proposed to stabilise the transmembrane helix bundle in active
conformations of ADGRG2 and ADGRG3²⁻⁵. In our ADGRL4 structure, three of the
four canonical UQC residues were conserved, including W6.53, which is stabilised by
a hydrogen bond interaction with Q7.49 as is observed in the canonical UQC motif.
The F5.43/F5.47 component of the UQC motif is absent in ADGRL4, as it is in other
aGPCRs such as ADGRF1, ADGRL3, and ADGRG5, suggesting that this position may
be dispensable or variable across aGPCR subfamilies. The conserved F $\phi\phi$ G in
ADGRL4 mediates the transmembrane helix 6 kink. Importantly, Q7.49, which
interacts with W6.53, is also closely packed against the F $\phi\phi$ G motif, implying an
analogous structural role in stabilizing TM6 conformation during activation. In the
H(N)L(M)Y motif of ADGRL4 (H3.53, L3.54, Y3.55), L3.54 directly contacts Y391 of
the $\alpha 5$ helix at the C-terminus of Gq, forming a hydrophobic interface with the G protein.
In summary, our results show that ADGRL4 contains all three 7TM activation motifs
with partially conserved spatial architecture, and with all three motifs being important
for receptor activation.

Within the **methods section** the following text has been added (line 743-759):

**Mutagenesis of putative 7TM activation motif regions.**

Alignment of 31 human protein-coding adhesion GPCRs was performed using
GPCRdb⁸. Protein-coding status was defined as per the HUGO Gene Nomenclature
Committee (HGNC)⁹, Ensembl¹⁰, and GeneCards¹¹ databases. ADGRF2 and
ADGRE4P were not included as both human genes are considered to be
pseudogenes. Scanning alanine mutagenesis was performed on eleven ADGRL4
amino acid residues within three putative 7TM activation motifs. Putative upper
quaternary core residues: F505^{3.44}, M508^{3.47}, W631^{6.53} (Q656^{7.49} also included as it
forms a hydrogen bond with W631^{6.53} in the canonical upper quaternary core motif);
putative P/F/W/L $\phi\phi$ G motif residues: F625^{6.47}, L626^{6.48}, L627^{6.49}, G628^{6.50}; putative

H(N)L(M)Y motif residues: H514^{3.53}, L515^{3.54} and Y516^{3.55}. Individual selected alanine
 mutations were cloned into versions of CTF1 (SmBiT tagged). Construct sequence
 fidelity was assessed using Sanger sequencing. G protein coupling assays were
 performed as described earlier with interactions between mutagenesis constructs
 (SmBiT tagged) and LgBiT-mini-G_q or LgBiT-mini-G_qΔα5 constructs determined by
 luminescence and normalised as previously described.

A new main figure has been added reporting the abovementioned results: (Fig.5; line 1111-
 1128)

**Fig.5 | Activation motifs in ADGRL4.** **a**, Amino acid residue conservation in aGPCRs
 of the upper quaternary core (UQC), FφφG motif and H(N)L(M)Y motif regions (φ refers
 a hydrophobic residue in this context): bold, conserved residues; italics, non-
 conserved residues; green, hydrophobic residues; blue, positively charged residues;
 red, negatively charged residues; yellow, polar uncharged residues. **b**, Alanine
 mutagenesis performed on amino acid residues within three putative 7TM activation
 motifs (UQC, FφφG motif and H(N)L(M)Y motif regions) with interactions between
 mutagenesis constructs (-SmBiT tagged) and LgBiT-mini-G_q or LgBiT-mini-G_qΔα5
 constructs determined by luminescence. Luminescence results were normalised by
 the expression levels of SmBiT- and LgBiT-tagged constructs. LgBiT-mini-G_qΔα5
 normalised luminescence samples were subtracted from LgBiT-mini-G_q samples, with
 results presented as a percentage of CTF1 G_q activity. Data represent three biological

replicates performed in duplicate; error bars represent the SEM (****, $p < 0.0001$). All
 data and significance values for graph **b** are shown in Extended Data Table 4. **c**,
 Structure of ADGRL4 (purple) with activation motif residues indicated: green, UQC
 motif; magenta, $F\phi\phi G$ motif; yellow, $H(N)L(M)Y$ motif. Q656^{7,49} (forming a hydrogen
 bond with W631^{6,53}) is listed in grey. Parts of H1 and H2 have been removed for clarity.
 The tethered agonist (cyan) and the C-terminus of G_q (orange) are also shown.

Two additional panels (**b,c**) have been added to Extended Data Fig.5 showing additional
 data relating to the abovementioned 7TM activation motif mutagenesis results: (Extended
 Data Fig.5; line 1247-1262)

Extended Data Fig.5 | Expanded tethered agonist and 7TM activation motifs
 alanine mutagenesis G_q complementation assays. **a**, Interactions between
 tethered agonist mutagenesis constructs (SmBiT tagged) and LgBiT-mini- G_q or LgBiT-
 mini- $G_q\Delta\alpha5$ constructs were determined by luminescence. The blue stars refer to
 comparisons to CTF1 + mini- G_q . Data represent three biological replicates, with each

assay containing 4 technical replicates; error bars represent the SEM (****, $p < 0.0001$;
*, $p < 0.05$; ns, non-significant). All data values and significance values are listed in
Extended Data Table 3. **b**, Cell surface expression levels of 7TM activation motif
mutagenesis constructs contrasted with CTF1, as determined by flow cytometry (α -HA).
Data represent six biological replicates performed in duplicate; error bars represent
the SEM. **c**, Interactions between 7TM activation motif mutagenesis constructs (SmBiT
tagged) and LgBiT-mini-G_q or LgBiT-mini-G_q $\Delta\alpha 5$ constructs as determined by
luminescence. The blue stars refer to comparisons to CTF1 + mini-G_q. Data represent
three biological replicates performed in duplicate; error bars represent the SEM (****,
$p < 0.0001$; ***, $p < 0.001$; **, $p < 0.01$; $p < 0.05$; ns, non-significant). All data values and
significance values are listed in Extended Data Table 3.

A new figure (Supplementary Figure 2) has been added reporting the aGPCR family-wide
 sequence alignment performed as a precursor to the 7TM activation motif experiments:
 (Supplementary Fig. 2; line 1281-1287)

 **Supplementary Fig. 2 | Expanded aGPCR amino acid alignment.** Alignment of 31
 human protein-coding aGPCRs showing transmembrane loops 3, 5 and 7, with
 Wootten numbering¹. 7TM activation motif regions for ADGR4 are highlighted with
 white boxes: upper quaternary core (UQC), FφφG and H(N)L(M)Y motifs (φ refers to
 hydrophobic residues). ADGRF2 and ADGRE4P were not included in the alignment as
 both human genes are considered to be pseudogenes.

MINOR POINTS

ORIGINAL REQUEST

Fig. 1

Generally, the authors should add their statistical result comparing FL1 to CTF1 and FLmut1
to CTF1 coupling to the intact G protein (just as annotated in Fig. 1d) in Fig. 1e-g even
though the results indicated no significant difference.

COMMENT

There is still one significance marker missing in Fig. 1d, i.e. between FLmut1 (grey) vs.
CTF1 (grey). Please add this to the figure.

Thank you for pointing this out. Figure 1d has now been updated and shows the previously
missing significance marker (****) for the comparison between FLmut1 (grey) and (CTF1)
grey.

The updated Fig.1d is pasted below: (line 1048-1064)

**Fig.1 | ADGRL4 couples weakly to mini-G_q.** **a**, Cartoon representation of ADGRL4
highlighting its extracellular domains and the activation mechanism. **b**, Cartoon
representation of ADGRL4 coupling constructs. **c**, Cartoon representation of N-
terminal LgBiT tethered mini-G protein or β-arrestin constructs. **d**, Interactions between
constructs of ADGRL4-SmBiT and LgBiT-mini-G_q (grey bars) were determined by
luminescence and compared to cells transfected with ADGRL4-SmBiT constructs and
LgBiT-mini-G_qΔα5 controls (white bars). Luminescence results were normalised by the

expression levels of SmBiT- and LgBiT-tagged constructs (Extended Data Figure 1).
Data represent three biological replicates, with each assay containing 3 technical
replicates; error bars represent the SEM (****, $p < 0.0001$; ***, $p < 0.001$; **, $p < 0.01$; *,
$p < 0.05$; ns, non-significant). **e-g**, Interactions between constructs of ADGRL4-SmBiT
and a variety of LgBiT-mini-G proteins (grey bars) or LgBiT-mini-G $\Delta\alpha 5$ control proteins
(white bars): **e**, mini-G_s; **f**, mini-G₁₂; **g**, mini-G_o. Data represents three biological
replicates; error bars represent the SEM. **h**, Interactions between constructs of
ADGRL4-SmBiT and either LgBiT- β arr1 (chequered bars) or LgBiT- β arr2 (lined bars).
All assays were performed three times in triplicate. All the data values and significance
values for graphs **d-h** are shown in Extended Data Table 1.

All other comments were satisfactorily addressed.

We thank the reviewer for the positive feedback.

**Reviewer #2 (Remarks to the Author):**

The authors have largely brushed off my concerns. They provided partially incomplete
explanations of revised text as an attempt to clarify issues raised by the critiques.

We thank the reviewer for their feedback and have completed the additional comparisons
requested.

Regarding the appendage of N-terminal tags to CTF-only versions of the adhesion GPCRs,
the authors chose to cherry-pick specific references that are attributed to one research lab,
and to ignore many other studies that were perhaps performed more rigorously, to support
their position that N-terminal tags do not impact ELTD-1 tethered agonism and signaling. An
issue with each of the referenced studies that parallels potential issues with the ELTD-1
constructs used in the current study is that there is no calibration of the signaling assays that
directly compared whether there might be much stronger signaling of naked N-terminal CTF
constructs versus the N-terminal tagged versions. Could the signaling assays with mini-G
proteins even support measuring a range of low efficacy activation to very high efficacy
versions? It might be best to perform the signaling measurements using an orthogonal assay
platform and to compare to the mini-G protein results. These issues are especially pertinent
to the current study because it claims to show weak G_q coupling and no G_o coupling
(although there appear to be statistically significant differences amongst the updated G_o
assays). What if the reason the signaling appears to be weak is because there is an
abnormal N-terminal tag blocking full access of the tethered agonist to the orthosteric site?

In light of your concern about the N-terminal HA tag, we have now investigated whether the
N-terminal tag has an effect on detectable G protein coupling using our NanoBiT assay. This
was performed by removing the N-terminal HA tag from the FL1, FLmut1 and CTF1 constructs
(all C-terminally tagged with SmBiT) and to generate Δ HA-FL1, Δ HA-FLmut1 and Δ HA-CTF1,
respectively, and comparing G protein recruitment to their HA-tagged versions.

Results show that for G_q coupling, Δ HA-FL1 produced a 221% increase in G_q coupling relative
to FL1 ($p=0.0003$); and that Δ HA-CTF1 produced a 270% increase in G_q coupling relative to
CTF1 ($p<0.0001$). There was no G_q coupling difference between Δ HA-FLmut1 and FLmut1
($p=0.9$). For G_o coupling, there was no difference between Δ HA-FL1 and FL1 ($p=0.2$), Δ HA-
FLmut1 and FLmut1 ($p=0.9$), and between Δ HA-CTF1 and CTF ($p=0.1$). These results are

presented in Extended Data Fig.1d-e, and in Extended Data Table 1. These data clearly show
 that the HA tag at the N-terminus, in these assays, affects G protein recruitment, as the
 reviewer intimated.

The following has been added to the manuscript:

- 1. Extended Data Fig.1d-e: G_q and G_o HA-tagged vs ΔHA G protein coupling
- experiments (line 1160-1194)
- 2. Extended Data Table 1: statistical information from the G_q and G_o HA-tagged vs ΔHA
- G protein coupling experiments (line 1147-1148 for the updated table)

Extended Data Fig.1 | Representative receptor-SmBiT, LgBiT-mini-G and β-
 arrestin expression levels (n=3). HA-tag cell surface expression levels as detected
 by flow cytometry. Constructs used as positive controls did not contain an HA-tag
 (HCRTR2, ADRB1, GPR35, APLNR). **b**, Whole cell expression of LgBiT-tethered mini-
 G/β-arrestin constructs as detected using Nano-Glo. **c**, Whole-cell expression of

SmBiT-tethered G protein coupling constructs as detected using Nano-Glo. All
ADGRL4 coupling constructs had similar whole-cell expression levels. The observed
differences in whole-cell expression levels between positive controls and ADGRL4
constructs was due to the use of distinct promoters: the positive control receptors were
expressed under a weak thymidine kinase (TK) promoter, whereas the ADGRL4
constructs were expressed under a strong cytomegalovirus (CMV) promoter. Although
the above displays results from a single representative biological replicate, all mini-G_q,
mini-G_s, mini-G₁₂, mini-G_o and β -arrestin related assays (cell-surface HA flow
cytometry, whole-cell mini-G levels, whole-cell receptor expression) were completed
as three biological replicates, with each assay containing 3 technical replicates. **d**,
Assessment of the effect of N-terminal HA tags on G_q coupling. Interactions between
aGPCR constructs (-SmBiT tagged) and LgBiT-mini-G_q or LgBiT-mini-G_q $\Delta\alpha$ 5
constructs were determined by luminescence. Luminescence results were normalised
by the expression levels of SmBiT- and LgBiT-tagged constructs. LgBiT-mini-G_q $\Delta\alpha$ 5
normalised luminescence samples were subtracted from LgBiT-mini-G_q samples, with
results presented as a percentage of HA-positive constructs. Δ HA versions represent
receptors without N-terminal HA tags. Data represent three biological replicates
performed in duplicate; error bars represent the SEM (****, $p < 0.0001$; ***, $p < 0.001$). All
the data values and significance values for graph **d** are shown in Extended Data Table
1 and in the source data file. **e**, Assessment of the effect of N-terminal HA tags on G_o
coupling. Interactions between aGPCR constructs (-SmBiT tagged) and LgBiT-mini-G_o
or LgBiT-mini-G_o $\Delta\alpha$ 5 constructs were determined by luminescence. Luminescence
results were normalised by the expression levels of SmBiT- and LgBiT-tagged
constructs. LgBiT-mini-G_o $\Delta\alpha$ 5 normalised luminescence samples were subtracted
from LgBiT-mini-G_o samples, with results presented as a percentage of HA-positive
constructs. Δ HA versions represent receptors without N-terminal HA tags. Data
represent three biological replicates performed in duplicate; error bars represent the
SEM. All the data values and significance values for graph **e** are shown in Extended
Data Table 1 and in the source data file.

The following has been added to the results section: (line 150-158)

As N-terminal tags may affect GPCR coupling and signalling, we investigated whether
N-terminal HA tags negatively affected our G protein coupling results. Constructs
without an N-terminal HA tag were therefore developed (DHA-FL1; DHA-FLmut1,
DHA-CTF1) and were assayed with the same NanoLuc assay. Results for mini-G_q
confirm our finding that ADGRL4 couples to G_q, albeit with lower efficiency when an

N-terminal HA-tag is present on the tethered agonist (Extended Data Fig.1d, Extended
Data Table 1). N-terminal HA tagging apparently had no effect on coupling with mini-
Go (Extended Data Fig.1e, Extended Data Table 1). As we required the N-terminal HA
tag for our structure pipeline, namely for detection during cell line development and
during purification, we continued to make use of HA-tagged constructs.

The following has been added to the methods section: (line 578-583)

Additional C-terminal SmBiT tagged constructs without an N-terminal tag were
generated: Δ HA-FL1; Δ HA-FLmut1; and Δ HA-CTF. Construct sequence fidelity was
assessed using Sanger sequencing. G protein coupling assays were performed as
described earlier with interactions between mutagenesis constructs (SmBiT tagged)
and LgBiT-mini-G_q (or LgBiT-mini-G_o) or LgBiT-mini-G_q $\Delta\alpha$ 5 (or LgBiT-mini-G_o $\Delta\alpha$ 5)
constructs determined by luminescence and normalised as previously described.

What if the reason the signaling appears to be weak is because there is an abnormal N-
terminal tag blocking full access of the tethered agonist to the orthosteric site?

This is of course a possibility. However, when we inspect the structure of ADGRL4 or any
other aGPCR, the N-terminus of the tethered agonist has direct access to what would be the
extracellular environment if the receptor was in the plasma membrane of a cell. We can
therefore see no impediment from a structural perspective that full engagement would not
occur. However, we cannot make any comment on the *rate* of tethered agonist engagement,
which we feel is more likely to be perturbed.

One other point we would like to clarify regarding these assays. We realise in hindsight that
the phraseology used in the methods of 'signalling assays' was inaccurate as we were
measuring G protein recruitment and not any downstream consequences. We have therefore
re-phrased 'signalling' to 'coupling'. We never meant to imply in the manuscript that we were
studying *signalling*, as this would entail far more detailed analyses in native cells and, although
very interesting, is beyond the scope of the current work.

The authors do see a pose of the tethered agonist within the orthosteric site that is
reminiscent of the poses of tethered agonists in other active state adhesion GPCRs, but
again it is concerning that the high resolution density of the tethered agonist happened to
stop precisely at the first authentic residue of the tethered agonist and there was no density
or resolved structure for the artificial tag residues that remained immediately N-terminal to

this portion of the receptor. Hand waving disclaimer statements that this region was just
unordered should not make this major issue just go away. Were the tags spontaneously
cleaved off right before the threonine residue? Could this be why the constructs signal and
no density for the tags appeared?

The purified ADGRL4 construct that was used for structure determination contained two major
components of apparent molecular weight on SDS-PAGE of 100 kDa and 80kDa (Extended
Figure 3h). The calculated molecular weights for the receptor is 32 kDa, the C-terminal mini-
G_q tag is 30 kDa and the N-terminal tag is 38 kDa. If all the N-terminal tag was cleaved off as
suggested by the reviewer, then the apparent molecular weight of the construct would be 62
405 kDa and is incompatible with the lowest molecular weight of the receptor observed on SDS-
406 PAGE (80 kDa). We acknowledge that there was some proteolysis of the construct, probably
from the N-terminal tag, but the SDS-PAGE data implies that a minimum of 8kDa (i.e. 70-80
amino acid residues) still exists at the N-terminus. Cryo-EM structure determination relies on
observing an *ordered* mass and cannot detect regions that are highly flexible, such as the N-
terminal tag we have used.

The authors state that the major goal of the research was to solve the structure of active-
state ELTD1, but they also declare that the signaling “is solved” for ELTD-1. The former is
clear, but the latter could be an overclaim based on the concerns noted about the N-terminal
tags.

We thank the reviewer for this comment. We have revised the paper to state that we
detected G protein coupling with G_q and have used this to determine the high resolution
structure of ADGRL4. We have removed any reference to solving the signalling for ADGRL4.

**Reviewer #3 (Remarks to the Author):**

The authors provide justified responses to my comments without providing any additional
experimental data. I would recommend at least including discussion of the limitations of the
structural insights and signaling assays using engineered G proteins.

As we discussed above for comments made by referee 2, we apologise for using the word
'*signalling*' in the context of this work. We measured G protein recruitment and did not study
any downstream consequences. We have therefore re-phrased '*signalling*' to '*coupling*'
throughout the manuscript.

We thank the reviewer for their comment and suggestion which we have completed in full,
adding a section in our discussion describing the limitations of the structural insights and
coupling assays using engineered G proteins. We have completely replaced the previous
limitations section with the new text below.

The following text has been added to the discussion section: (line 454-486)

In terms of limitations, we used engineered mini-G chimeras for measuring the
recruitment of G protein to the receptor, and used a C-terminally tethered mini-G_q for
structure determination. Mini-G proteins lack the G_α helical domain and have been
shown to form high-affinity, stable nucleotide-insensitive receptor complexes for
structural biology applications^{12,13}. Our mini-G_q coupling sensor was a G_s/G_q chimera¹³
in which residues in the α5 helix were mutated to match G_q. This has the potential to
subtly alter coupling in comparison to wild type G proteins.

Another potential limitation is that the NanoBiT G protein recruitment assay reports
proximity between SmBiT-tagged receptor and LgBiT-tagged mini-G rather than
assessing GDP-GTP exchange, GTP hydrolysis or downstream second-messenger
production, and it was performed using overexpressed ADGRL4 in HEK293T cells.
The dominant G_q result derives from a constitutively active CTF with markedly lower
surface expression than full-length constructs. We detected no robust coupling to other
transducers in this background, although a very weak trend with mini-G_o was observed.
We acknowledge that determination of physiologically relevant G protein coupling must
be demonstrated in native endothelial cells or tumours. The intention of this paper was
to determine an active-state structure of ADGRL4, and we needed to have a stable
receptor-G protein complex to facilitate this work.

Finally, our conclusions on the structure and G protein recruitment assays should be
viewed in the context of the requirements for ADGRL4 expression, purification and
structure determination. The structure represents an active-state C-terminal fragment
(CTF2B) of ADGRL4 bound to a C-terminally tethered mini-G_q chimera that recruits
βγ, assembled and analysed in LMNG/CHS detergent micelles with N-terminal tags
for purification and detection retained. These choices were necessary to stabilise the
weakly coupled ADGRL4-G_q complex *in vitro* and to facilitate purification and alignment
during cryo-EM image processing, but they could conceivably bias receptor-G protein
geometry and orientation, omit potential membrane-lipid allostery, and mask
alternative or transient states. Single-particle cryo-EM emphasises the most populated
conformation, so it remains possible that low-occupancy active-state conformations or
dynamic states exist that were not recovered by our classification as our analysis relied
on averaging the most frequent 3D-conformations.

Regarding cancer involvement, as I mentioned, it is often possible to find references
suggesting direct or indirect associations with cancer for most GPCRs. However, such
associations alone do not necessarily justify considering a GPCR as a valid therapeutic
target for cancer treatment. On the other hand, I am not strongly against the title since it is
not a major issue.

We thank the reviewer for their comment.

**References:**

- 1 Wootten, D., Simms, J., Miller, L. J., Christopoulos, A. & Sexton, P. M. Polar
transmembrane interactions drive formation of ligand-specific and signal pathway-
biased family B G protein-coupled receptor conformations. *Proceedings of the*
*National Academy of Sciences of the United States of America* **110**, 5211-5216
(2013). <https://doi.org/10.1073/pnas.1221585110>
- 2 Ping, Y.-Q. *et al.* Structures of the glucocorticoid-bound adhesion receptor GPR97–
Go complex. *Nature* **589**, 620-626 (2021). [https://doi.org/10.1038/s41586-020-](https://doi.org/10.1038/s41586-020-03083-w)
[03083-w](https://doi.org/10.1038/s41586-020-03083-w)
- 3 Ping, Y.-Q. *et al.* Structural basis for the tethered peptide activation of adhesion
GPCRs. *Nature* (2022). <https://doi.org/10.1038/s41586-022-04619-y>
- 4 Qu, X. *et al.* Structural basis of tethered agonism of the adhesion GPCRs ADGRD1
and ADGRF1. *Nature* (2022). <https://doi.org/10.1038/s41586-022-04580-w>
- 5 Xiao, P. *et al.* Tethered peptide activation mechanism of the adhesion GPCRs
ADGRG2 and ADGRG4. *Nature* (2022). <https://doi.org/10.1038/s41586-022-04590-8>
- 6 Zhu, X. *et al.* Structural basis of adhesion GPCR GPR110 activation by stalk peptide
and G-proteins coupling. *Nature communications* **13**, 5513 (2022).
<https://doi.org/10.1038/s41467-022-33173-4>
- 7 Seufert, F., Chung, Y. K., Hildebrand, P. W. & Langenhan, T. 7TM domain structures
of adhesion GPCRs: what's new and what's missing? *Trends in biochemical sciences*
**48**, 726-739 (2023). <https://doi.org/10.1016/j.tibs.2023.05.007>
- 8 Herrera, L. P. T. *et al.* GPCRdb in 2025: adding odorant receptors, data mapper,
structure similarity search and models of physiological ligand complexes. *Nucleic*
*acids research* **53**, D425-d435 (2025). <https://doi.org/10.1093/nar/gkae1065>
- 9 Seal, R. L. *et al.* Genenames.org: the HGNC resources in 2023. *Nucleic acids*
*research* **51**, D1003-D1009 (2022). <https://doi.org/10.1093/nar/gkac888>
- 10 Dyer, S. C. *et al.* Ensembl 2025. *Nucleic acids research* **53**, D948-d957 (2025).
<https://doi.org/10.1093/nar/gkae1071>
- 11 Stelzer, G. *et al.* The GeneCards Suite: From Gene Data Mining to Disease Genome
Sequence Analyses. *Curr Protoc Bioinformatics* **54**, 1.30.31-31.30.33 (2016).
<https://doi.org/10.1002/cpbi.5>
- 12 Carpenter, B. & Tate, C. G. Engineering a minimal G protein to facilitate
crystallisation of G protein-coupled receptors in their active conformation. *Protein*
*Eng Des Sel* **29**, 583-594 (2016). <https://doi.org/10.1093/protein/gzw049>
- 13 Nehmé, R. *et al.* Mini-G proteins: Novel tools for studying GPCRs in their active
conformation. *PloS one* **12**, e0175642 (2017).
<https://doi.org/10.1371/journal.pone.0175642>

**Reviewer #1 (Remarks to the Author):**

This is the second revision of a manuscript reporting the G protein bound structure of L4-
CTF and its function as a Gq coupled receptor. Most of my comments were now all
satisfactorily addressed and I recommend acceptance of the paper.

We thank the reviewer for recommending acceptance.

MAJOR POINTS

ORIGINAL REQUEST

The now available L4 signalling assay lends itself to clarify the presence or absence of
previously identified activation motifs within the 7TM domain of other aGPCRs (i.e. the upper
quaternary core, P/F/W/LφφG, H(N)L(M)Y, reviewed in 10.1016/j.tibs.2023.05.007). The
authors are asked first determine whether any of the previously characterised signaling
motifs are present in the 7TM domain of L4. If so, they should mutagenise them and test
them for changes in Gq coupling.

**COMMENT:** The authors have now fulfilled my initial request to analyse the role of the
identified 7TM activation motifs and the tethered agonist sequence in full. I congratulate
them for the meticulous analyses they have undertaken to bind their findings on the L4
structure to its activation by its tethered agonist. In particular to see that mutating the TA
core sequence from the +2 position onwards profoundly compromises the activity of the
receptor is reassuring and will support the function of L4 as a G protein coupled receptor,
which was previously disputed. Also addressing the concern of the potentially harmful effect
of the N-terminal HA-tag, that was raised by another reviewer, was important and underlines
the findings in the coupling assays.

The results will me most useful for others to compare their molecular pharmacological and
structural findings with and learn from the L4 receptor.

Thank you very much for the positive feedback.

MINOR POINTS

ORIGINAL REQUEST

There is still one significance marker missing in Fig. 1d, i.e. between FLmut1 (grey) vs.
CTF1 (grey). Please add this to the figure.

**COMMENT:** This has been resolved now.

Thank you for the confirmation.

**Reviewer #2 (Remarks to the Author):**

The authors have experimentally addressed the critiques and provided adequate
explanations and interpretations of their data.

We thank the reviewer for their comments and suggestions. We have completed all
requests.

I think it is much more than nuance to demonstrate that the new tested constructs with
authentic tethered agonists provide higher apparent efficacies than the HA-tagged
constructs. This should have been examined previously during the peer review of the major
papers that the current authors primarily used for the design of their original submission
constructs. (First revision paraphrased response: "N-terminal tags did not affect
coupling/signaling in references, X, Y, Z"). The tags do affect coupling/signaling/recruitment
and it is good that the revised manuscript will help to rectify this for the field. The authors
might consider moving this new data from Extended Data Figure (1) into the main
manuscript figure set. It's not convenient to do this in a twice revised manuscript, but it would
be best not to bury this in the ED.

In response to the reviewer request we have moved the two relevant panels from **Extended**
Data Fig. 1 d-e, to Fig. 1 i-j.

**Fig. 1 | ADGRL4 couples weakly to mini-G_q.** **a**, Cartoon representation of ADGRL4
 highlighting its extracellular domains and the activation mechanism. **b**, Cartoon
 representation of ADGRL4 coupling constructs. **c**, Cartoon representation of N-
 terminal LgBiT tethered mini-G protein or β-arrestin constructs. **d**, Interactions between
 constructs of ADGRL4-SmBiT and LgBiT-mini-G_q (grey bars) were determined by
 luminescence and compared to cells transfected with ADGRL4-SmBiT constructs and
 LgBiT-mini-G_qΔα5 controls (white bars). Luminescence results were normalised by the
 expression levels of SmBiT- and LgBiT-tagged constructs (Supplementary Figure 1).
 Data represent three biological replicates, with each assay containing three technical
 replicates; all comparisons were tested using an unpaired two-tailed Student's t-test;
 error bars represent the SEM (****, $p < 0.0001$; ***, $p < 0.001$; **, $p < 0.01$; *, $p < 0.05$; ns,
 non-significant). **e-g**, Interactions between constructs of ADGRL4-SmBiT and a variety
 of LgBiT-mini-G proteins (grey bars) or LgBiT-mini-GΔα5 control proteins (white bars):
 **e**, mini-G_s; **f**, mini-G₁₂; **g**, mini-G_o. Data represents three biological replicates with each
 assay containing three technical replicates; all comparisons were tested using an
 unpaired two-tailed Student's t-test; error bars represent the SEM. **h**, Interactions
 between constructs of ADGRL4-SmBiT and either LgBiT-βarr1 (chequered bars) or

LgBiT- β arr2 (lined bars). Data represents three biological replicates with each assay
containing three technical replicates; all comparisons were tested using an unpaired
two-tailed Student's t-test; error bars represent the SEM (***, $p < 0.001$; *, $p < 0.05$; ns,
non-significant). **i**, Assessment of the effect of N-terminal HA tags on G_q coupling.
Interactions between aGPCR constructs (-SmBiT tagged) and LgBiT-mini- G_q or LgBiT-
mini- $G_q\Delta\alpha 5$ constructs were determined by luminescence. Luminescence results were
normalised by the expression levels of SmBiT- and LgBiT-tagged constructs. LgBiT-
mini- $G_q\Delta\alpha 5$ normalised luminescence samples were subtracted from LgBiT-mini- G_q
samples, with results presented as a percentage of HA-positive constructs. Δ HA
versions represent receptors without N-terminal HA tags. Data represent three
biological replicates, with each assay containing two technical replicates; all
comparisons were tested using an unpaired two-tailed Student's t-test; error bars
represent the SEM (****, $p < 0.0001$; ***, $p < 0.001$; **, $p < 0.01$; *, $p < 0.05$; ns, non-
significant). **j**, Assessment of the effect of N-terminal HA tags on G_o coupling.
Interactions between aGPCR constructs (-SmBiT tagged) and LgBiT-mini- G_o or LgBiT-
mini- $G_o\Delta\alpha 5$ constructs were determined by luminescence. Luminescence results were
normalised by the expression levels of SmBiT- and LgBiT-tagged constructs. LgBiT-
mini- $G_o\Delta\alpha 5$ normalised luminescence samples were subtracted from LgBiT-mini- G_o
samples, with results presented as a percentage of HA-positive constructs. Δ HA
versions represent receptors without N-terminal HA tags. Data represent three
biological replicates, with each assay containing two technical replicates; all
comparisons were tested using an unpaired two-tailed Student's t-test; error bars
represent the SEM (****, $p < 0.0001$; ***, $p < 0.001$; **, $p < 0.01$; *, $p < 0.05$; ns, non-
significant). All data and significance values for graphs **d-j** are shown in Supplementary
Table 1.

The response to the absence of ordered data for the HA tag in the structure is hand waving.
If there is no data, there is no data, but if the HA tagged tethered agonist binds to the
orthosteric site in a less efficient manner than the authentic tethered agonist (not tested), it
could be that the structure does not represent a fully-active 7TM state and therefore could
explain the weakly engaged pose that G_q exhibited. I think the authors should at least
acknowledge this possibility with a sentence in the discussion.

We have added the following text in the discussion (lines 463-472) with regards to the
above:

“A potential criticism that can be made of the ADGRL4 structure is that the presence
of the N-terminal tags has affected the structure in some way. We do not think this is
the case given the close similarity in position of the tethered ligand in ADGRL4 with
other aGPCRs and also the positions of conserved amino acid residue side chains in
the conserved motifs of aGPCRs. Of course, structures present a static view of a
protein in a thermodynamically favourable state in the conditions used for structure
determination and will not yield information regarding, for example, the kinetics of
activation. Thus the N-terminal tags have a demonstrable effect on the ability of
ADGRL4 to recruit G_q (Fig. 1i), yet the structure is of relevance in understanding how
ADGRL4 functions.”

Otherwise, the manuscript is improved.

We thank the reviewer for their positive feedback.

**Reviewer #3 (Remarks to the Author):**

The authors have provided additional mutagenesis data on the tethered agonist and key
7TM activation motifs of the receptor, which substantially strengthen their structural insights.
The additional discussion addressing the limitations of the study is also satisfactory and
resolves my earlier concerns.

We thank the reviewer for their positive feedback.

I have only one minor suggestion: please update the label of ADGRL4 to ADGRL4-CTF2B in
the main Figure 2,

As requested, we have updated the text label ADGRL4 to ADGRL4-CTF2B within Fig. 2 and
within the associated legend:

“Fig. 2 | 3.14Å Cryo-EM structure of active-state ADGRL4-CTF2B. a-b, Cryo-EM
density map (a) and structure model (b) for active-state ADGRL4-CTF2B in complex
with mini-G_q and βγ. The tethered agonist is highlighted in turquoise with the insert
showing an alternative magnified view. “

and replace G_q with mini-G_q in the main Figure 6, along with updates to the figure legends
for accuracy.

As requested, we have updated the Figure 6 (changing G_q to mini-G_q) and have updated the
figure legend accordingly:

b Receptor contacts to mini-G_q

c Mini-G_q contacts to receptors

	ICL1	239	242	243	246	353	354	357	358	ICL2	557	561	563	564	565	567	568	ICL3	637	638	640	641	644	645	847	849	856
ADGRL4	-	R	-	-	R	L	M	L	V	F	E	E	M	H	T	-	-	K	P	S	C	-	G	-	Q	K	K
ADGRL3	R	-	-	R	-	L	M	L	V	F	E	E	M	H	T	-	-	K	P	S	C	-	G	-	Q	K	K
ADGRE5	-	-	-	R	-	L	M	L	V	F	Q	Q	L	-	-	-	-	-	-	-	-	-	G	-	Q	K	K
ADGRF1	-	K	T	S	R	L	-	R	I	V	F	-	V	L	R	-	-	-	-	-	-	-	K	-	T	L	D

	Q35	HN52	HN102	R38	HN103	L41	H412	R342	V358	S601	C359	S603	F351	F376	H508	H512	K380	H513	D381	H514	I382	H515	E83	H516	L984	H517	Q385	H519	N387	H520	L388	H522	E390	H523	Y391	N392	H524	Y391	H525	L393	V394	beta	R52	D312	Total							
ADGRL4	-	-	-	-	-	-	-	-	-	-	-	-	-	-	-	-	-	-	-	-	-	-	-	-	-	-	-	-	-	-	-	-	-	-	-	-	-	-	-	-	-	-	-	-	-	-	-	-	-	-	20	
ADGRL3	2	1	-	1	1	1	1	5	1	2	-	2	-	2	-	-	-	-	-	-	3	-	3	1	1	1	1	3	1	1	3	1	2	3	1	3	1	4	1	6	4	4	3	3	3	3	3	3	3	3	3	37
ADGRE5	-	3	1	-	-	-	-	1	-	-	-	-	2	1	3	-	-	-	-	-	3	-	3	3	1	1	1	1	1	1	1	1	-	-	-	-	1	2	3	3	3	3	3	3	3	3	3	3	3	29		
ADGRF1	-	-	-	-	-	-	-	-	-	-	-	-	2	-	-	-	-	-	-	-	-	-	-	-	-	-	-	-	-	-	-	-	-	-	-	-	-	-	-	-	-	-	-	-	-	-	-	-	-	28		

Fig. 6 | Comparison of receptor-mini-G_q interfaces. **a**, Interactions between ADGRL4 (turquoise) or ADGRL3 (yellow; PDB ID: 7wy5) with the α-subunit (orange) and the β-subunit (dark green) of the mini-G_q heterotrimeric G protein are depicted. Structures are shown as cartoons with residues at the interfaces depicted as sticks (red, oxygen atoms; dark blue, nitrogen atoms). Dashes represent interatomic interactions: yellow, van der Waals interactions (distance ≤ 3.9 Å); red, potential polar contacts. **b**, Comparison of the amino acid contacts (distance cut-off ≤ 3.9Å) made by the receptor to mini-G_q for active-state ADGRL4, ADGRL3, ADGRE5 and ADGRF1; green, hydrophobic residues; blue, positively charged residues; red, negatively charged residues; yellow, polar uncharged residues. PDB IDs: ADGRL3-mini-G_q, 7wy5; ADGRE5-mini-G_q, 7ydm; and ADGRF1-mini-G_q, 7wxu. **c**, Comparison between the amino acid contacts made by mini-G_q to receptor for active-state ADGRL4, ADGRL3, ADGRE5 and ADGRF1. Residue colouring scheme as before; contact frequency coloured separately.
